# Are Language Models Actually Useful for Time Series Forecasting?

**Mingtian Tan**
University of Virginia
wtd3gz@virginia.edu

**Mike A. Merrill**
University of Washington
mikeam@cs.washington.edu

**Vinayak Gupta**
University of Washington
vinayak@cs.washington.edu

**Tim Althoff**
University of Washington
althoff@cs.washington

**Thomas Hartvigsen**
University of Virginia
hartvigsen@virginia.edu

## Abstract

Large language models (LLMs) are being applied to time series forecasting. But are language models actually useful for time series? In a series of ablation studies on three recent and popular LLM-based time series forecasting methods, we find that removing the LLM component or replacing it with a basic attention layer does not degrade forecasting performance—in most cases, the results even improve! We also find that despite their significant computational cost, pretrained LLMs do no better than models trained from scratch, do not represent the sequential dependencies in time series, and do not assist in few-shot settings. Additionally, we explore time series encoders and find that patching and attention structures perform similarly to LLM-based forecasters.[1]

## 1 Introduction

Time series analysis is a critical problem across many domains, including disease propagation forecasting [8], retail sales analysis [3], healthcare [26, 17] and finance [31]. A great deal of recent work in time series analysis (constituting repositories with more than 1200 total stars on GitHub) has focused on adapting pretrained large language models (LLMs) to classify, forecast, and detect anomalies in time series [15, 50, 22, 4, 5, 32, 14, 44, 16]. These papers posit that language models, being advanced models for sequential dependencies in text, may generalize to the sequential dependencies in time series data. This hypothesis is unsurprising given language models are now pervasive in machine learning research. However, direct connections between language modeling and time series forecasting remain largely undefined. So to what extent is language modeling *really* beneficial for traditional time series tasks?

Our claim is simple but profound: **popular LLM-based time series forecasters perform the same or worse than basic LLM-free ablations, yet require orders of magnitude more compute**. Derived from extensive ablations, this reveals a worrying trend in contemporary time series forecasting literature. Our goal is not to imply that language models will never be useful for time series. In fact, recent works point to many exciting and promising ways that language and time series interact, like time series reasoning [25, 7, 45, 42, 37], social understanding [6] and financial reasoning [36, 20]. Rather, we aim to highlight surprising findings that existing methods do very little to use the innate reasoning power of pretrained language models on established time series tasks.

We substantiate our claim by performing three ablations of three popular and recent LLM-based forecasting methods [50, 15, 22] using eight standard benchmark datasets from reference methods

---

[1]All resources needed to reproduce our work are available: `https://github.com/BennyTMT/LLMsForTimeSeries`

and another five datasets from MONASH [13]. First, we successfully reproduce results from the original publications. Then, we show that replacing language models with simple attention layers, basic transformer blocks, randomly-initialized language models, and *even removing the language model entirely*, yields comparable or better performance. The same performance was observed on another five datasets that were not studied by the reference methods.

Next, we compare the training and inference speed of these methods against their ablations, showing that these simpler methods reduce training and inference time by up to three orders of magnitude while maintaining comparable performance. Then, to investigate the source of LLM forecaster's performance, we further explore time series encoders. We find that a simple linear model with an encoder composed of patching and attention can achieve forecasting performance similar to that of LLMs. Next, we test whether the sequence modeling capabilities of LLMs transfer to time series by shuffling input time series and find no appreciable change in performance. Finally, we show that LLMs do not even help forecasting in few-shot settings with 10% of the training data. We discuss the implications of our findings and suggest that time series methods that use large language models are better left to multimodal applications [4, 12, 38] that require textual reasoning.

The key contributions we make in this paper are as follows:

- We propose three straightforward ablation methods for methods that pass time series into LLMs for forecasting. We then ablate three top-tier methods on thirteen standard datasets and find that LLMs fail to convincingly improve time series forecasting. However, they significantly increase computational costs in both training and inference.

- We study the impact of an LLM's pretraining by re-initializing their weights prior to forecasting. We find that this has no impact on forecasting performance. Additionally, in shuffling input time series, we find no evidence the LLMs successfully transfer sequence modeling abilities from text to time series and no indication they help in few-shot settings.

- We find a very simple model, with patching and attention as encoder, can achieve performance similar to LLMs. This suggests a massive gap between the benefits LLMs pose and the time series forecasting problem, despite a rapid rush to adopt LLMs.

## 2   Related Work

Here, we summarize the key works relevant to LLM-based time series models. They can be broadly classified into three sections: (i) time series forecasting using LLMs; (ii) encoders in LLM time series models; and (iii) smaller and efficient neural models for time-series.

**Time Series Forecasting Using LLMs.** Recently, with the development of Large Language Models (LLMs) [10, 29, 34] and their demonstrated multi-modal capabilities, more researchers have successfully applied LLMs to time series forecasting tasks [14, 16, 5, 4]. Chang *et al.,* [5] used finetuning the transformer module and positional encoding in GPT-2 to align pre-trained LLMs with time series data for forecasting tasks. Zhou et al. [50] proposed a similar finetuning method, named "OneFitsAll", for time series forecasting with GPT-2. Additionally, Jin et al. [15] introduced a reprogramming method to align LLM's Word Embedding with time series embeddings, showing good representation of time series data on LLaMA [34]. Similarly, CALF [22] and TEST [32] adapted word embeddings to enable LLMs to forecast time series data effectively. In addition to time-series forecasting models, Liu et al. [23] show that these models can be extended to classifying health-time series, such as heart-rate and daily-footsteps. These models have also been shown to outperform supervised neural models in few-shot settings.

**Encoders in LLM Time Series Models.** In order for an LLM to learn from text it must first be discretized and encoded as word tokens which are $1 \times d$ vectors [10, 29, 34]. Similarly, LLM-based methods for time series learn discrete time series tokens. One method is to segment the time series into overlapping patches, which effectively shortens the time series while retaining its features [15, 50, 5, 4, 28, 27, 11]. Other methods decompose time series into trend, seasonal components, and residual components [4, 28]. Lastly, Liu et al. [22] feed the multivariate time series using a Transformer to enable different channels to learn the dynamics of other channels. These embedding procedures are followed by a linear neural network layer that projects the time series encoding to the same dimensions used by the pre-trained LLM.

| Dataset | ETTh1 & ETTh2 | ETTm1 & ETTm2 | Traffic | Electricity | Weather | Illness |
|---|---|---|---|---|---|---|
| Channels | 7 | 7 | 862 | 321 | 21 | 7 |
| Sampling-Rate | 1 Hour | 15 Min. | 1 Hour | 1 Hour | 10 Min. | 1 Week |
| Timesteps | 17,420 | 69,680 | 17,544 | 26,304 | 52,696 | 966 |

Table 1: Statistics for all datasets used in reference methods [50, 22, 15].

| Method | Base Model | Learnable LM Parameters | Positional Embeddings | Align Word Embeddings | Multimodal |
|---|---|---|---|---|---|
| OneFitsAll [50] | GPT-2 | Add&Norm | Fine-Tune | ○ | ○ |
| Time-LLM [15] | LLaMA | None | Freeze | ● | ● |
| CALF [22] | GPT-2 | LoRA | Fine-Tune | ● | ○ |

Table 2: Three popular methods for time series forecasting with Large Language Models.

**Small and Efficient Neural Forecasters.** In addition to LLMs, there has been a large body of research on smaller yet efficient frameworks that outperform their bulky counterparts in time series forecasting [19, 47, 33, 24, 2]. For example, Zeng et al. [46] present DLinear, an incredibly simple model that combines decomposition techniques and achieves better forecasting performance than state-of-the-art transformer-based time series architectures at the time, such as Informer [48], FED-former [49], and Autoformer [40]. Furthermore, Xu et al. [43] introduces a lightweight model with only 10k parameters, which captures both amplitude and phase information in the time-series to outperform transformer-based models.

## 3 Experimental Setup

We use three state-of-the-art methods for time series forecasting and propose three ablation methods for LLMs: (i) "**w/o LLM**"; (ii) "**LLM2Attn**"; (iii) and "**LLM2Trsf**". To evaluate the effectiveness of LLMs in time series forecasting, we test these methods on eight standard datasets.

### 3.1 Reference Methods for Language Models and Time Series

We experiment with three recent methods for time series forecasting using LLMs. All models were published between December 2023 and May 2024 and are popular, with their GitHub repositories collectively amassing 1,245 stars. These methods are summarized in Table 2, and use either GPT-2 [29] or LLaMA [34] as base models, with different alignment and fine-tuning strategies.

- **OneFitsAll [50]**: OneFitsAll, sometimes called GPT4TS, applies instance norm and patching to the input time series and then feeds it into a linear layer to obtain a input representation for the language model. The multi-head attention and feed forward layers of the language model are frozen while the positional embeddings and layer norm are optimized during training. A final linear layer is used to transform the language model's final hidden states into a prediction.

- **Time-LLM [15]**: In Time-LLM the input time series is tokenized via patching and aligned with a low-dimensional representation of word embeddings using multi-head attention. The outputs of this alignment, combined with the embeddings of descriptive statistical features, are passed to a frozen pre-trained language model. The output representations of the language model are then flattened and passed through a linear layer to obtain a forecast.

- **CALF [22]**: CALF embeds the input time series by treating each channel as a token. One half of the architecture is a "textual branch" which uses cross attention to align the time series representation with a low dimensional representation of the language model's word embeddings. This representation is then passed through a pretrained, frozen language model to obtain a "textual prediction". Simultaneously, a "temporal" branch learns a low-rank adapter for a pretrained language model based on the input time series to produce a "temporal prediction" which is used for inference. The model includes additional loss terms that enforce similarity between these representations.

**Reproducibility Note.** While experimenting with each model, we tried to replicate the conditions of their original papers. We used the original hyper-parameters, runtime environments, and code, including model architectures, training loops, and data-loaders. To ensure a fair comparison, we have included error metrics from the original papers alongside our results wherever possible.

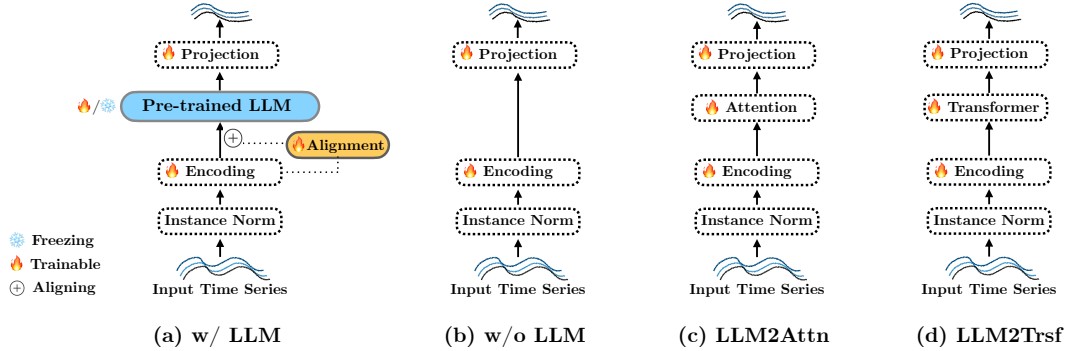

Figure 1: Overview of all LLM ablation methods. Figure (a) represents time series forecasting using an LLM as the base model. In some works, the LLM components are frozen [15, 14], while in others, they undergo fine-tuning [50, 22, 4]. Figure (b) shows the model with the LLM components removed, retaining only the remaining structure. Figure (c) replaces the LLM components with a single-layer self-attention mechanism. Figure (d) replaces the LLM components with a simple Transformer.

## 3.2 Proposed Ablations

To isolate the influence of the LLM in an LLM-based forecaster, we propose three ablations: removing the LLM component or replacing it with a simple block. Specifically, for each of the three methods we make the following three modifications:

- **w/o LLM** (Figure 1 (b)). We remove the language model entirely, instead passing the input tokens directly to the reference method's final layer.
- **LLM2Attn** (Figure 1 (c)). We replace the language model with a single randomly-initialized multi-head attention layer.
- **LLM2Trsf** (Figure 1 (d)). We replace the language model with a single randomly-initialized transformer block.

In the above ablations, we keep left parts of the forecasters unchanged (trainable). For example, as shown in Figure 1 (a), after removing the LLM, the input encodings are passed directly to the output projection. Alternatively, as shown in Figure 1 (b) or (c), after replacing the LLM with attention or a transformer, they are trained along with the remaining structure of the original method.

## 3.3 Datasets and Evaluation Metrics

**Benchmark Datasets.** We evaluate on the following real-world datasets: (1) **ETT** [21]: encompasses seven factors related to electricity transformers across four subsets: ETTh1 and ETTh2, which have hourly recordings, and ETTm1 and ETTm2, which have recordings every 15 minutes; (2) **Illness** [40]: includes the weekly recorded influenza illness among patients from the Centers for Disease Control, which describes the ratio of patients seen with influenza-like illness to the total number of patients; (3) **Weather** [40]: local climate data from 1,600 U.S. locations, between 2010 and 2013, and each data point consists of 11 climate features; (4) **Traffic** [40]: is an hourly dataset from California transportation department, and consists of road occupancy rates measured on San Francisco Bay area freeways; (5) **Electricity** [35]: contains the hourly electricity consumption of 321 customers from 2012 to 2014. The train-val-test split for ETT datasets is 60%-20%-20%, and for Illness, Weather, and Electricity datasets is 70%-10%-20% respectively. The statistics for all datasets is given in Table 1. We highlight that these datasets, with the same splits and size, have been extensively used to evaluate time-series forecasting ability of LLM-based and other neural models for time-series data [48, 50, 4, 15, 5, 46, 40, 49]. (6) **Exchange Rate** [18]: collected between 1990 and 2016, it contains daily exchange rates for the currencies of eight countries (Australia, British, Canada, Switzerland, China, Japan, New Zealand and Singapore). (7) **Covid Deaths** [13]: contains daily statistics of COVID-19 deaths in 266 countries and states between January and August 2020. (8) **Taxi (30 min)** [1]: contains taxi rides from 1,214 locations in New York City between January 2015 and January 2016. The data is collected every 30 minutes, with an average of 1,478 samples. (9) **NN5 (Daily)** [13]: contains daily cash withdrawal data from 111 ATMs in the UK, with each ATM having

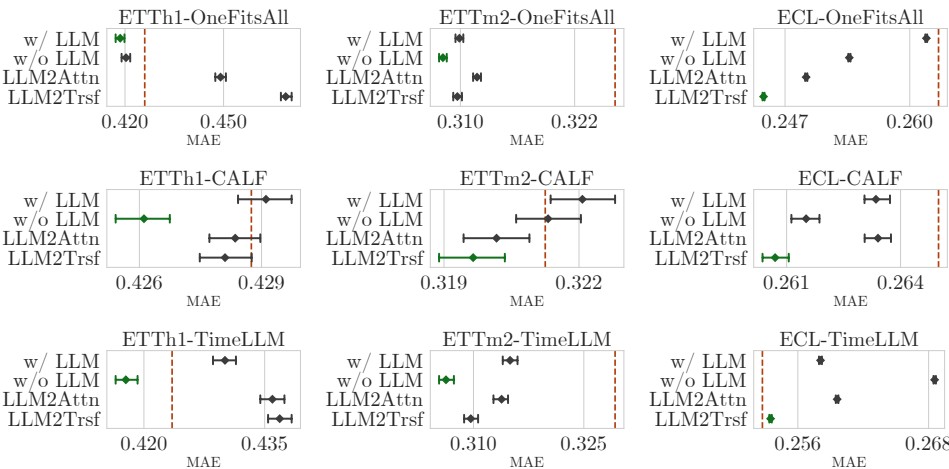

Figure 2: In the above examples, only OneFitsAll "w/ LLM" performs better than the ablation methods on ETTh1, but there is substantial overlap in bootstraped confidence intervals. The figures show the comparison of OneFitsAll, CALF, and Time-LLM using LLMs and ablations (*i.e.*, **w/o LLM**, **LLM2Attn**, and **LLM2Trsf**) on ETTh1, ETTm2, and Electricity, and the vertical dashed lines represent the results from the original work. Others Figures for MSE and other datasets are available in Figure 5 and Figure 6 in the Appendix.

791 data points. (10) **FRED-MD** [13]: contains 107 monthly macroeconomic indices released by the Federal Reserve Bank since 01/01/1959. It was extracted from the FRED-MD database.

**Evaluation Metrics and Setup.** We report the results in terms of mean absolute error (MAE) and mean squared error (MSE) between predicted and true values of the time-series. Mathematically, given a test-set with $\mathcal{D}$ elements, $\text{MAE} = \frac{1}{|\mathcal{D}|}\sum_{t_i \in \mathcal{D}}[|c_i - \widehat{c}_i|]$ and $\text{MSE} = \frac{1}{|\mathcal{D}|}\sum_{t_i \in \mathcal{D}}(c_i - \widehat{c}_i)^2$, where $c_i$ and $\widehat{c}_i$ denote the true value and predicted value at the $i$-th index of the time-series respectively.

## 4 Results

In this section, we provide the details of our comprehensive evaluation of all baseline LLM models for time-series forecasting. Specifically, we ask the following research questions. **(RQ1)** Do pretrained language models contribute to forecasting performance? **(RQ2)** Are LLM-based methods worth the computational cost? **(RQ3)** Does language model pretraining help performance on forecasting tasks? **(RQ4)** Do LLMs represent sequential dependencies in time series? **(RQ5)** Do LLMs help with few-shot learning? **(RQ6)** Where does the performance come from?

### 4.1 Do pretrained language models contribute to forecasting performance? (RQ1)

Our results show that pretrained LLMs **are not useful for time series forecasting tasks yet**. Overall, as shown in Table 3, across 13 datasets and two metrics, ablations out perform Time-LLM methods in 26/26 cases, CALF in 22/26 cases, and OneFitsAll in 19/26 cases. We averaged results over different predicting lengths, as in [50, 15, 22]. Across all prediction lengths (thirteen datasets and four prediction lengths) ablations outperformed Time-LLM, CALF, and OneFitsAll in 35/40, 31/40, and 29/40 cases as measured by MAE, respectively. To ensure a fair comparison, we also report results from each method's original paper alongside our replication. For specific results refer to Appendix E.1. To better evaluate the effectiveness of LLMs and ablation methods, we include 95% bootstrapped confidence intervals for each task. In tasks where LLMs performed better, such as OneFitsAll with ETTh1, shown in Figure 2, there is still substantial overlap in the confidence intervals with the ablation method "**w/o LLM**" in MAE. Other datasets results and MSE metrics are shown in Figure 5 and Figure 6 in the Appendix. To summarize, our results on the evaluation above, it is hard to conclude that LLMs are effective in time series forecasting.

### 4.2 Are LLM-based methods worth the computational cost? (RQ2)

In the previous section, we showed that LLMs do not meaningfully improve performance on time series forecasting tasks. Here, we evaluate the computational intensity of these methods with their

**Time-LLM [15]**

| Model → | Time-LLM | | w/o LLM | | LLM2Attn | | LLM2Trsf | | From Original Paper | |
|---|---|---|---|---|---|---|---|---|---|---|
| Dataset ↓ | MAE | MSE | MAE | MSE | MAE | MSE | MAE | MSE | MAE | MSE |
| ETTh1 | 0.432 | 0.417 | **0.419** | **0.405** | 0.437 | 0.422 | 0.439 | 0.429 | 0.423 | 0.408 |
| ETTh2 | 0.396 | 0.360 | **0.383** | **0.345** | 0.389 | 0.353 | 0.394 | 0.359 | 0.383 | 0.334 |
| ETTm1 | 0.377 | 0.356 | **0.371** | **0.350** | 0.376 | 0.356 | 0.377 | 0.359 | 0.371 | 0.329 |
| ETTm2 | 0.315 | 0.260 | **0.307** | **0.252** | 0.314 | 0.259 | 0.310 | 0.253 | 0.329 | 0.250 |
| Illness | 0.894 | 2.017 | 0.924 | 1.956 | 0.849 | **1.789** | **0.837** | 1.795 | 0.801 | 1.435 |
| Weather | 0.270 | 0.243 | 0.272 | 0.243 | 0.254 | **0.224** | **0.254** | 0.226 | 0.257 | 0.225 |
| Traffic | 0.281 | 0.421 | 0.295 | 0.428 | 0.276 | 0.416 | **0.275** | **0.416** | 0.263 | 0.387 |
| Electricity | 0.259 | 0.164 | 0.269 | 0.171 | 0.260 | 0.167 | **0.254** | **0.161** | 0.252 | 0.158 |
| Exchange Rate | 0.448 | 0.422 | **0.413** | **0.384** | 0.432 | 0.403 | 0.442 | 0.422 | - | - |
| Covid Deaths | 0.089 | 0.189 | 0.080 | 0.198 | 0.058 | 0.086 | **0.054** | **0.079** | - | - |
| Taxi (30 Min) | 0.277 | 0.163 | 0.286 | 0.176 | 0.269 | 0.157 | **0.255** | **0.141** | - | - |
| NN5 (Daily) | 0.432 | 0.402 | 0.425 | 0.379 | 0.411 | 0.364 | **0.401** | **0.347** | - | - |
| FRED-MD | 0.0004 | 5e-7 | **0.0002** | **3e-7** | 0.0046 | 2.53e-5 | 0.0008 | 2.6e-6 | - | - |
| **# Wins** | 0 | | 12 | | 2 | | 12 | | - | |
| **#Parameters** | 6651.82M | | 0.55M | | 0.55M | | 0.66M | | - | |

**CALF [22]**

| Model → | CALF | | w/o LLM | | LLM2Attn | | LLM2Trsf | | From Original Paper | |
|---|---|---|---|---|---|---|---|---|---|---|
| Dataset ↓ | MAE | MSE | MAE | MSE | MAE | MSE | MAE | MSE | MAE | MSE |
| ETTh1 | 0.431 | 0.431 | **0.428** | 0.436 | 0.430 | **0.428** | 0.430 | 0.430 | 0.428 | 0.432 |
| ETTh2 | 0.383 | 0.351 | 0.383 | 0.352 | **0.382** | **0.349** | 0.383 | 0.350 | 0.382 | 0.349 |
| ETTm1 | 0.391 | 0.396 | 0.390 | 0.397 | 0.390 | 0.396 | **0.390** | **0.394** | 0.390 | 0.395 |
| ETTm2 | 0.323 | 0.283 | 0.322 | 0.282 | 0.321 | **0.281** | **0.320** | 0.281 | 0.321 | 0.281 |
| Illness | 0.869 | 1.699 | 0.861 | 1.639 | 0.892 | 1.748 | **0.860** | **1.630** | - | - |
| Weather | **0.273** | **0.251** | 0.277 | 0.257 | 0.279 | 0.258 | 0.277 | 0.255 | 0.274 | 0.250 |
| Traffic | 0.284 | 0.443 | 0.278 | 0.439 | 0.275 | 0.430 | **0.271** | **0.426** | 0.281 | 0.439 |
| Electricity | 0.266 | 0.175 | 0.262 | 0.174 | 0.264 | 0.175 | **0.261** | **0.172** | 0.265 | 0.175 |
| Exchange Rate | 0.417 | 0.388 | 0.409 | **0.367** | 0.417 | 0.389 | **0.409** | 0.367 | - | - |
| Covid Deaths | 0.084 | 0.163 | **0.066** | 0.115 | 0.131 | 0.431 | 0.066 | **0.106** | - | - |
| Taxi (30 Min) | **0.258** | **0.142** | 0.264 | 0.147 | 0.267 | 0.150 | 0.267 | 0.150 | - | - |
| NN5 (Daily) | 0.403 | 0.362 | **0.386** | **0.336** | 0.463 | 0.445 | 0.415 | 0.381 | - | - |
| FRED-MD | 0.0012 | 2.9e-6 | **0.0011** | **2.7e-6** | 0.0015 | 4.9e-6 | 0.0017 | 4.5e-6 | - | - |
| **# Wins** | 4 | | 7 | | 4 | | 11 | | - | |
| **#Parameters** | 180.25M | | 8.17M | | 10.5M | | 13.68M | | - | |

**OneFitsAll [50]**

| Model → | OneFitsAll | | w/o LLM | | LLM2Attn | | LLM2Trsf | | From Original Paper | |
|---|---|---|---|---|---|---|---|---|---|---|
| Dataset ↓ | MAE | MSE | MAE | MSE | MAE | MSE | MAE | MSE | MAE | MSE |
| ETTh1 | **0.420** | 0.417 | 0.422 | **0.417** | 0.452 | 0.465 | 0.474 | 0.525 | 0.426 | 0.427 |
| ETTh2 | **0.388** | **0.353** | 0.389 | 0.356 | 0.402 | 0.375 | 0.397 | 0.367 | 0.394 | 0.354 |
| ETTm1 | 0.377 | 0.362 | **0.374** | **0.358** | 0.379 | 0.369 | 0.379 | 0.369 | 0.383 | 0.351 |
| ETTm2 | 0.310 | **0.253** | 0.309 | 0.255 | 0.312 | 0.256 | 0.310 | 0.254 | 0.326 | 0.266 |
| Illness | 0.852 | 1.871 | 0.924 | 1.960 | **0.829** | **1.763** | 0.850 | 1.830 | 0.903 | 1.925 |
| Weather | **0.254** | 0.226 | 0.272 | 0.245 | 0.256 | **0.226** | 0.256 | 0.228 | 0.270 | 0.236 |
| Traffic | 0.273 | 0.420 | 0.273 | 0.439 | 0.266 | 0.415 | **0.256** | **0.409** | 0.294 | 0.414 |
| Electricity | 0.262 | 0.169 | 0.254 | 0.165 | 0.250 | 0.162 | **0.245** | **0.157** | 0.263 | 0.166 |
| Exchange Rate | 0.378 | 0.357 | **0.361** | **0.323** | 0.376 | 0.350 | 0.393 | 0.387 | - | - |
| Covid Deaths | 0.057 | 0.075 | **0.050** | **0.073** | 0.058 | 0.103 | 0.078 | 0.162 | - | - |
| Taxi (30 Min) | **0.252** | **0.138** | 0.259 | 0.143 | 0.259 | 0.145 | 0.257 | 0.140 | - | - |
| NN5 (Daily) | 0.438 | 0.438 | 0.422 | **0.385** | 0.423 | 0.390 | **0.420** | 0.386 | - | - |
| FRED-MD | 0.0006 | 1.2e-6 | **0.0002** | **4e-7** | 0.0006 | 1.5e-6 | 0.0012 | 2.4e-6 | - | - |
| **# Wins** | 7 | | 11 | | 3 | | 5 | | - | |
| **#Parameters** | 91.36M | | 9.38M | | 10.71M | | 13.54M | | - | |

Table 3: Forecasting performance of all models – Time-LLM, CALF, and OneFitsAll and results from our ablations. All results are averaged across different prediction lengths, though full results are available in Appendix E.1. Results in Red denote the best-performing model. # Wins refers to the number of times the method performed best, and # Params is the number of model parameters. "-" means the dataset is not included in the original paper.

nominal performance in mind. The language models in our reference methods use hundreds of millions and sometimes billions of parameters to perform time series forecasting. Even when the parameters of the language models are frozen they still contribute to substantial overhead during training and inference. For instance, Time-LLM has 6642 M parameters and takes 3003 minutes to train on the Weather dataset whereas ablation methods have only 0.245 M parameters and take 2.17 minutes on average. Information about training other methods on ETTh1 and Weather datasets are shown in Table 4. In the case of inference time, we divide by the maximum batch size to give an estimate of inference time per example. Time-LLM, OneFitsAll, and CALF take, on average, **28.2**,

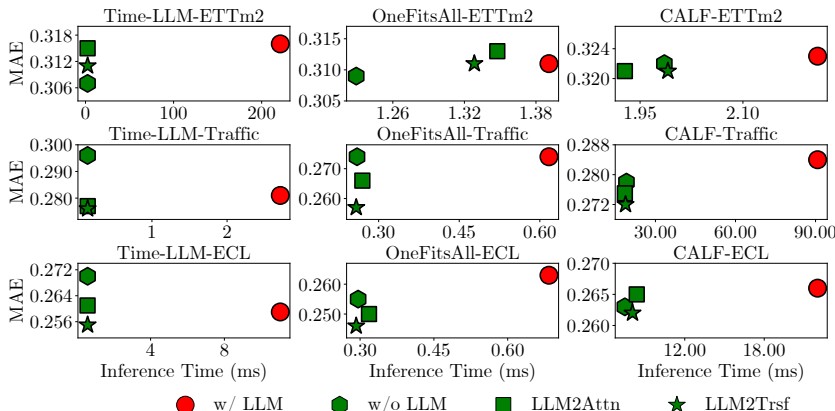

Figure 3: Ablation methods consume less time for inference while providing better forecasting performance. The figure above shows the inference time and prediction accuracy of Time-LLM, OneFitsAll, and CALF on ETTm2, Traffic, and Electricity datasets, averaged across prediction lengths. For more datasets and MSE metrics refer to Figure 7 and Figure 8 in the Appendix.

|  | Method | Time-LLM (LLaMA) | | OneFitsAll (GPT-2) | | CALF (GPT-2) | |
|---|---|---|---|---|---|---|---|
|  |  | # Param (M) | Time (min) | # Param (M) | Time (min) | # Param (M) | Time (min) |
| ETTh1 | w/ LLM | 6652 | 181 | 85 | 7.36 | 180 | 3.28 |
|  | w/o LLM | 0.198 | 0.99 | 3 | 0.27 | 8 | 0.35 |
|  | LLM2Attn | 0.202 | 1.41 | 5 | 0.70 | 10 | 0.37 |
|  | LLM2Trsf | 0.336 | 0.84 | 8 | 0.64 | 13 | 0.40 |
| Weather | w/ LLM | 6642 | 3003 | 86 | 152 | 180 | 12 |
|  | w/o LLM | 0.198 | 1.91 | 4 | 16 | 8 | 2.32 |
|  | LLM2Attn | 0.202 | 2.22 | 7 | 21 | 10 | 2.14 |
|  | LLM2Trsf | 0.336 | 2.38 | 10 | 24 | 13 | 1.89 |

Table 4: In time series tasks, LLM (LLaMA and GPT-2) significantly increases training time. The table shows the number of model parameters (in millions) and total training time (in minutes) for three methods predicting over a length of 96 on ETTh1 and Weather data. Compared with original method "**w/ LLM**" are "**w/o LLM**", "**LLM2Attn**" and "**LLM2Trsf**".

**2.3**, and **1.2** times longer than the modified models. Examples can be seen in Figure 3, where the green marks (ablation methods) are typically below the red one (LLM) and are positioned towards the left of the axis, indicating a lower computational costs and better forecasting performance. Other datasets and MSE metric refer to Figure 7 and Figure 8 in Appendix. In conclusion, the computational intensity of LLMs in time series forecasting tasks does not result in a corresponding performance improvement.

### 4.3 Does language model pretraining help performance on forecasting tasks? (RQ3)

Our evaluation in this section indicates that **pretraining with language datasets is unnecessary for time series forecasting.** To test whether the knowledge learned during pretraining meaningfully improves forecasting performance we experimented with different combinations of pretraining and finetuning CALF's [22] language model on time series.

- **Pretrain + Finetune (Pre+FT)**. This is the original method, wherein a pretrained language model is finetuned on time series data. In the case of CALF, the base language model is frozen and low rank adapters (LoRA) are learned.

- **Random Initialization + Finetune (woPre+FT)**. Does the textual knowledge from pretraining aid time series forecasting? In this method we randomly initialize the weights of the language model (thereby erasing any effect of pretraining) and train the LLM from scratch.

- **Pretrain + No Finetuning (Pre+woFT)**. How much does finetuning on time series improve prediction performance? For this baseline we again leave the language model frozen and forgo

| Methods | Pre+FT (GPT-2) | | woPre+FT | | Pre+woFT | | woPre+woFT | |
|---|---|---|---|---|---|---|---|---|
| | MAE | MSE | MAE | MSE | MAE | MSE | MAE | MSE |
| ETTh1 | 0.4312 | **0.4313** | 0.4284 | 0.4362 | **0.4267** | 0.4342 | 0.4365 | 0.4474 |
| ETTh2 | 0.3838 | 0.3510 | 0.3839 | **0.3508** | **0.3830** | 0.3514 | 0.3872 | 0.3554 |
| ETTm1 | 0.3910 | 0.3963 | 0.3933 | 0.4013 | **0.3898** | **0.3954** | 0.3949 | 0.4028 |
| ETTm2 | 0.3230 | 0.2831 | **0.3221** | 0.2852 | 0.3221 | **0.2827** | 0.3224 | 0.2829 |
| Illness | 0.8691 | 1.6996 | **0.8523** | **1.6146** | 0.8742 | 1.6640 | 0.8663 | 1.6381 |
| Weather | **0.2737** | **0.2510** | 0.2760 | 0.2520 | 0.2771 | 0.2535 | 0.2776 | 0.2582 |
| Traffic | 0.2844 | 0.4438 | **0.2771** | **0.4409** | 0.2820 | 0.4446 | 0.2863 | 0.4483 |
| Electricity | 0.2660 | 0.1758 | **0.2597** | **0.1669** | 0.2635 | 0.1730 | 0.2663 | 0.1784 |
| **# Wins:** | 3 | | 8 | | 5 | | 0 | |

Table 5: Randomly initializing LLM parameters and training from scratch (woPre) achieved better results than using a pretrained (Pre) model. "woFT" and "FT" refer to whether the LLM parameters are frozen or trainable.

| Dataset | ETTh1 | | | | Illness | | | |
|---|---|---|---|---|---|---|---|---|
| Input Ablation | Sf-all. | Sf-half. | Ex-half | Masking | Sf-all. | Sf-half. | Ex-half | Masking |
| Time-LLM | 51.8% | 5.6% | 79.6% | 32.5% | 99.0% | 33.6% | 34.9% | 64.6% |
| **w/o LLM** | 56.0% | 4.5% | 89.7% | 39.5% | 76.5% | 20.9% | 18.4% | 53.0% |
| **LLM2Attn** | 53.8% | 3.3% | 92.5% | 33.8% | 72.7% | 20.4% | 13.1% | 44.6% |
| **LLM2Trsf** | 50.3% | 3.4% | 89.2% | 34.8% | 74.5% | 23.0% | 14.3% | 49.3% |
| OneFitsAll | 62.1% | 6.1% | 16.6% | 31.3% | 86.2% | 30.9% | 36.7% | 77.5% |
| **w/o LLM** | 58.6% | 6.1% | 19.2% | 36.1% | 68.9% | 13.0% | 17.3% | 43.5% |
| **LLM2Attn** | 68.5% | 9.0% | 15.0% | 34.4% | 108.3% | 39.8% | 44.2% | 74.2% |
| **LLM2Trsf** | 58.0% | 7.8% | 12.6% | 30.2% | 90.8% | 27.4% | 40.3% | 60.6% |
| CALF | 50.5% | 9.6% | 5.6% | 8.5% | 113.0% | 47.4% | 24.4% | 22.9% |
| **w/o LLM** | 56.2% | 12.1% | 6.1% | 10.4% | 118.0% | 50.4% | 45.8% | 28.9% |
| **LLM2Attn** | 51.9% | 10.8% | 5.8% | 7.3% | 87.3% | 42.4% | 35.1% | 25.8% |
| **LLM2Trsf** | 50.3% | 8.5% | 5.5% | 7.0% | 102.6% | 56.2% | 32.6% | 26.0% |

Table 6: For the input shuffling/masking experiments on ETTh1 (predict length is 96) and Illness (predict length is 24), the impact of shuffling the input on the degradation of time series forecasting performance does not change significantly before and after model modifications. Results of other predict lengths refer to table 21 in Appendix.

learning LoRAs. Results from this model are therefore indicative of the base language model's performance without additional guidance on processing time series.

- **Random Initialization + No Finetuning (woPre+woFT)**. This baseline is effectively a random projection from the input time series to a forecasting prediction and serves as a baseline comparison with the other methods.

Overall, as shown in Table 5, across 8 datasets using MAE and MSE metrics, the "Pretraining + Finetune" method performed the best 3 times, while "Random Initialization + Finetune" achieved this 8 times. This indicates that language knowledge offers very limited help for forecasting. However, "Pretrain + No Finetuning" and the baseline "Random Initialization + No Finetuning" performed the best 5 times and 0 times, respectively, suggesting that Language knowledge does not contribute meaningfully during the finetuning process. Detailed results refer to Table 20 in Appendix.

In summary, textual knowledge from pretraining provides very limited aids for time series forecasting.

### 4.4 Do LLMs represent sequential dependencies in time series? (RQ4)

Most time series forecasting methods that use LLMs finetune the positional encoding to help understand the position of time steps in the sequence [4, 50, 22, 5, 32]. We would expect a time series model with good positional representations to show a significant drop in predictive performance when the input is shuffled [46]. We applied three types of shuffling to the time series: shuffling the entire sequence randomly ("sf-all"), shuffling only the first half of the sequence ("sf-half"), and swapping the first and second halves of the sequence ("ex-half"). As shown in Table 6, **LLM-based methods were no more vulnerable to input shuffling than their ablations.** This implies that LLMs do not have unique capabilities for representing sequential dependencies in time series.

| Model | LLaMA | | w/o LLM | | LLM2Attn | | LLM2Trsf | |
|---|---|---|---|---|---|---|---|---|
| | MAE | MSE | MAE | MSE | MAE | MSE | MAE | MSE |
| ETTh1 | **0.522** | **0.555** | 0.543 | 0.621 | 0.547 | 0.611 | 0.566 | 0.656 |
| ETTh2 | **0.394** | **0.371** | 0.432 | 0.407 | 0.437 | 0.416 | 0.433 | 0.412 |
| ETTm1 | 0.426 | 0.404 | **0.403** | **0.393** | 0.428 | 0.440 | 0.438 | 0.457 |
| ETTm2 | 0.323 | 0.277 | **0.319** | **0.269** | 0.355 | 0.316 | 0.351 | 0.311 |
| Weather | 0.273 | **0.234** | **0.271** | 0.241 | 0.275 | 0.241 | 0.271 | 0.239 |
| Traffic | 0.306 | **0.429** | 0.303 | 0.431 | 0.302 | 0.432 | **0.298** | 0.432 |
| Electricity | **0.270** | **0.175** | 0.275 | **0.175** | 0.273 | 0.179 | 0.274 | 0.181 |
| # Wins: | 8 | | 7 | | 0 | | 1 | |

Table 7: In few-shot scenarios (10% dataset), LLaMA (Time-LLM) performs similarly to the ablation methods. LLaMA and "**w/o LLM**" each outperformed the other 8 times. Note that the results of Time-LLM is from the original paper [15].

| Model | GPT-2 | | w/o LLM | | LLM2Attn | | LLM2Trsf | |
|---|---|---|---|---|---|---|---|---|
| | MAE | MSE | MAE | MSE | MAE | MSE | MAE | MSE |
| ETTh1 | 0.543 | 0.649 | **0.542** | **0.642** | 0.545 | 0.646 | 0.544 | 0.643 |
| ETTh2 | 0.433 | 0.434 | **0.429** | **0.427** | 0.433 | 0.431 | 0.432 | 0.434 |
| ETTm1 | 0.500 | 0.574 | **0.499** | **0.572** | 0.503 | 0.581 | 0.507 | 0.586 |
| ETTm2 | 0.339 | 0.304 | **0.338** | **0.303** | 0.340 | 0.305 | 0.340 | 0.305 |
| Weather | **0.286** | **0.263** | 0.287 | 0.265 | 0.293 | 0.270 | 0.286 | 0.264 |
| Traffic | 0.369 | 0.571 | 0.337 | 0.517 | 0.341 | 0.518 | **0.333** | **0.510** |
| Electricity | 0.301 | 0.220 | **0.287** | **0.205** | 0.292 | 0.208 | 0.290 | 0.206 |
| # Wins: | 2 | | 10 | | 0 | | 2 | |

Table 8: In few-shot scenarios (10% dataset), Ablation methods perform much better than GPT-2 (CALF). Without LLMs, 12 out of 14 cases showed better performance.

### 4.5 Do LLMs help with few-shot learning in forecasting? (RQ5)

In this section, our evaluation demonstrates that LLMs are still **not meaningfully useful in few-shot learning scenarios.**

While our results indicate that LLMs are not useful for time series forecasting, it is nonetheless possible that knowledge encoded in pretrained weights could help performance in few-shot settings where data are scarce. To evaluate whether this is the case we trained models and their ablations on 10% of each dataset. Specifically, we evaluated LLaMA in Time-LLM methods. The results for LLaMA, shown in Table 7, compared LLaMA with completely removing the LLM (**w/o LLM**). There was no difference, with each performing better in 8 cases. We conducted similar experiments with CALF, a GPT-2-based method. Our results in Table 8 indicate that our ablations can perform better than LLMs in few-shot scenarios.

### 4.6 Where does the performance come from? (RQ6)

In this section, we evaluate common encoding techniques used in LLM time series models. We find that combining **patching with one-layer attention is a simple and effective choice.**

In subsection 3.2, we found that simple ablations of LLM-based methods did not decrease performance. To understand why such simple methods work so well we selected some popular techniques used for encoding in LLM time series tasks, such as patching [50, 15, 5, 32, 22, 11], decomposition [4, 28]. A basic transformer block also can be used to aid in encoding [22].

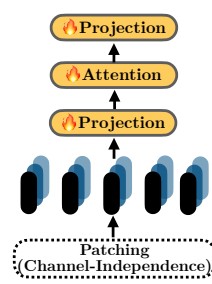

Figure 4: PAttn Model.

The specific results, shown in Table 18 in the Appendix, indicate that a structure combining patching and attention, named "PAttn", performs better than most other encoding methods on small datasets (with time stamps less than 1 million) and is even comparable to LLM methods. Its detailed structure, as shown in Figure 4, involves applying "instance norm" to the time series, followed by patching and projection. Then, one-layer

attention enables feature learning between patches. For larger datasets, such as Traffic (~15 million) and Electricity (~8 million), a model named "LTrsf," using the encoder from CALF [22], performs better. In those methods, finally, time series embedding will be projected with a single linear layer to forecast. Details of other encoders is in Appendix subsection D.3.

Overall, patching plays a crucial role in encoding. Additionally, basic Attention and Transformer blocks also effectively aid in encoding.

## 5 Conclusion

In this paper we showed that despite the recent popularity of LLMs in time series forecasting they do not appear to meaningfully improve performance. We experimented with simple ablations, showing that they maintain or improve the performance of the LLM-based counterparts while requiring considerably less compute. Once more, our goal is not to suggest that LLMs have no place in time series analysis. To do so would likely prove to be a shortsighted claim. Rather, we suggest that the community should dedicate more focus to the exciting tasks could be unlocked by LLMs at the interface of time series and language such as time series reasoning [25, 7, 45, 37], or social understanding [6].

## 6 Acknowledgements

We thank the University of Virginia's Research Computing team for maintaining the excellent high-performance computing resources that allowed us to conduct this research. This research was supported in part by NSF IIS-1901386, NSF CAREER IIS-2142794, NIH R01MH125179, the Bill & Melinda Gates Foundation (INV-004841), the Office of Naval Research (#N00014-21-1-2154), and the National Security Data & Policy Institute, Contracting Activity #2024-24070100001.

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

# Are Language Models Actually Useful for
# Time Series Forecasting?
# (Appendix)

## A    Limitations

Here, we discuss the limitations of our paper.

1. We evaluate the ability of LLMs using time-series forecasting. However, to get a better picture of how LLMs can work with time-series, this ability should be evaluated across other downstream tasks as well, such as time-series classification and question-answering.

2. Our evaluation is limited to only time-series datasets, *i.e.*, sequences with even time-intervals. However, there also exists a large fraction of data in the form of non-uniform series, such as payment records, online purchases, etc. Understanding the ability of LLMs in forecasting non-uniform sequences is also necessary to verify our claim on the usefulness of LLMs for time-series data.

## B    Broader Societal Impact

One of the major impacts our study will have is on the influx of models that use LLMs for modeling time-series. Our results will help researchers to not simply follow the trend of using LLMs in all applications, but to check their usability in detail. Specifically, these findings will help them determine if the LLM component is necessary and if the computational costs are reasonable for the specific setting. In addition to the research community, our findings on the better performance of smaller and simpler models will help develop scalable models that are easy to understand, interpret, and can be deployed cheaply in real-world applications. While we agree that a majority of our results are experimental and limited to selected datasets, we feel that these results will also help researchers narrow down their search space for better models in time-series forecasting, and not simply neglect the simpler models.

## C    License

All our contributions will be released under the MIT License.

## D    Additional Experimental Details

### D.1    System Configuration

We train and evaluate each reference method and each architecture modification using the same device. For Time-LLM [15], applying LlaMA-7B [34], we use NVIDIA A100 GPU with 80GB memory. For other methods [50, 22], applying GPT-2 [29], we use NVIDIA RTX A6000 GPU with 48GB memory. Though an analysis of memory footprint is beyond scope of this research, we note that training the baselines in the absence of LLM can be done within smaller GPUs as well.

### D.2    Baseline Hyper-Parameter Details

When reproducing the reference methods, we used the original repository's hyper-parameters and model structures. In the ablation study, due to the smaller model parameters, we adjusted the learning rate or increased the batch size in some cases. All other training details remained identical with the reference methods. The specific training details and hyper-parameters are provided in the code and run scripts of the repository[2]. Note that the training process and hyper-parameters for the simple methods can also be accessed via this link.

### D.3    Details of Encoder Exploration and Simple Methods

To investigate the source of LLM method performance, we conducted further research on encoders. We used various encoders to encode time series data, followed by a linear layer to project the time

---

[2]https://github.com/BennyTMT/LLMsForTimeSeries

series embeddings to the forecast. The encoder structure combining patching and attention is shown in Figure 4. The key difference between PAttn and PatchTST [27] is the absence of position embedding and the Feed Forward in PatchTST [27], or, more specifically, replacing the Transformer Encoder with a simple single-layer Attention structure. It could be a simple yet effective method to help evaluate the trade-off between cost and performance for newly proposed methods.

In addition, we propose three different neural models (i) "LTrsf": which performs better on larger datasets, uses CALF 's [22] encoder without cross-modal attention. This could be a potential source of CALF 's performance; (ii) "D-LTrsf"; and (iii) "D-PAttn": both of them decompose the time series into three sub-sequences and forecast each using the above two methods respectively, and then linearly combine the results for the final forecast. Across our results in Table 18, we note that even simpler models significantly outperform LLM-based time-series models. In detail, the LLM-based models, all combined we able to appear 33 times as the best and the second-best performer. However, "PAttn" was outperforming them by appearing 34 times as the best and the second-best performer. In addition, the comparison between "PAttn", "LTrsf" and other state-of-the-art methods is shown in Table 19.

# E    Additional Experiments

Here we present the results of additional experiments that also highlight the ability of LLMs in modeling time-series data.

## E.1    Confidence Intervals for Forecasting

Since LLMs and deep learning models in general are probabilistic in nature, their predictions can vary across different runs and different random initializations. Thus, we report the confidence intervals (CIs) for all MAE and MSE predictions made by our baseline models. We report the CIs for MAE prediction by Time-LLM, CALF, OneFitsAll in Tables 9, 11, and 13, for MSE predictions in Tables 10, 12, and 14, respectively. Across all results, we note that the range of variation is quite small, and these intervals do not affect our observations. To illustrate the subtle differences more clearly, we present the visualized results in Figure 5 and Figure 6.

**Confidence Intervals for other Datasets.**  To evaluate the generality of the ablations in the paper, we introduce five additional datasets that have not been studied by the reference methods [50, 15, 22]. The above datasets are used in many time series forecasting studies [30, 9, 39, 1, 46]. The prediction lengths for the "Exchange Rate" are "96, 192, 336, 720", as in [46, 18]. The prediction lengths for the other four datasets are 30, 48, 56, and 12, respectively, following the settings in Chronos [1]. As shown in Tables 15, 16, and  17, the forecasting performance on the five new datasets, using the three methods [50, 15, 22] we referenced, still demonstrates that language models are unnecessary for forecasting tasks.

## E.2    Complete Results

Here we provide the results for all methods and datasets that we were unable to add to the main paper.

**Inference Times.**  All results regarding "inference time" and forecast performance are shown in Figure 7 and Figure 8 respectively.

**Randomized Parameters and Random-Shuffling of Inputs.** The results of the randomized parameters are shown in Table 20. The remaining results for shuffled input are shown in Table 21,22,23 and 24.

| Models Dataset | Window | Time-LLM MAE | CI | w/o LLM MAE | CI | LLM2Attn MAE | CI | LLM2Trsf MAE | CI | From Paper |
|---|---|---|---|---|---|---|---|---|---|---|
| ETTh1 | 96 | 0.402 | (0.399,0.405) | 0.385 | (0.383,0.388) | 0.400 | (0.397,0.402) | 0.402 | (0.399,0.405) | 0.392 |
| | 192 | 0.421 | (0.418,0.424) | 0.412 | (0.409,0.415) | 0.423 | (0.420,0.426) | 0.426 | (0.424,0.429) | 0.418 |
| | 336 | 0.438 | (0.435,0.440) | 0.431 | (0.428,0.433) | 0.475 | (0.472,0.478) | 0.449 | (0.447,0.453) | 0.427 |
| | 720 | 0.468 | (0.465,0.470) | 0.452 | (0.450,0.455) | 0.452 | (0.449,0.455) | 0.480 | (0.477,0.483) | 0.457 |
| ETTh2 | 96 | 0.346 | (0.343,0.350) | 0.331 | (0.328,0.334) | 0.338 | (0.334,0.342) | 0.346 | (0.342,0.350) | 0.328 |
| | 192 | 0.391 | (0.387,0.396) | 0.374 | (0.369,0.378) | 0.384 | (0.381,0.389) | 0.390 | (0.386,0.393) | 0.375 |
| | 336 | 0.414 | (0.410,0.418) | 0.407 | (0.402,0.410) | 0.406 | (0.402,0.410) | 0.404 | (0.400,0.408) | 0.409 |
| | 720 | 0.434 | (0.429,0.437) | 0.424 | (0.421,0.428) | 0.430 | (0.426,0.434) | 0.437 | (0.434,0.441) | 0.420 |
| ETTm1 | 96 | 0.341 | (0.340,0.342) | 0.337 | (0.336,0.338) | 0.333 | (0.332,0.335) | 0.336 | (0.335,0.338) | 0.334 |
| | 192 | 0.369 | (0.368,0.371) | 0.360 | (0.359,0.361) | 0.366 | (0.364,0.367) | 0.366 | (0.365,0.368) | 0.358 |
| | 336 | 0.379 | (0.378,0.381) | 0.379 | (0.378,0.381) | 0.386 | (0.385,0.387) | 0.386 | (0.385,0.388) | 0.384 |
| | 720 | 0.419 | (0.418,0.420) | 0.410 | (0.409,0.411) | 0.422 | (0.421,0.423) | 0.419 | (0.418,0.421) | 0.411 |
| ETTm2 | 96 | 0.248 | (0.247,0.250) | 0.246 | (0.245,0.248) | 0.249 | (0.248,0.251) | 0.249 | (0.248,0.251) | 0.253 |
| | 192 | 0.304 | (0.303,0.306) | 0.285 | (0.283,0.287) | 0.293 | (0.291,0.295) | 0.294 | (0.292,0.295) | 0.293 |
| | 336 | 0.329 | (0.328,0.331) | 0.321 | (0.319,0.323) | 0.333 | (0.331,0.335) | 0.323 | (0.321,0.325) | 0.392 |
| | 720 | 0.382 | (0.380,0.385) | 0.377 | (0.375,0.379) | 0.384 | (0.382,0.387) | 0.377 | (0.376,0.380) | 0.379 |
| Illness | 24 | 0.807 | (0.772,0.841) | 0.913 | (0.879,0.950) | 0.848 | (0.812,0.884) | 0.837 | (0.805,0.870) | 0.727 |
| | 36 | 0.833 | (0.804,0.861) | 0.902 | (0.878,0.931) | 0.846 | (0.813,0.882) | 0.846 | (0.816,0.872) | 0.814 |
| | 48 | 1.012 | (0.986,1.041) | 0.932 | (0.907,0.958) | 0.828 | (0.805,0.847) | 0.805 | (0.785,0.842) | 0.807 |
| | 60 | 0.925 | (0.898,0.953) | 0.949 | (0.920,0.979) | 0.873 | (0.846,0.905) | 0.862 | (0.836,0.893) | 0.857 |
| Weather | 96 | 0.199 | (0.198,0.200) | 0.213 | (0.212,0.214) | 0.189 | (0.188,0.190) | 0.189 | (0.188,0.190) | 0.201 |
| | 192 | 0.261 | (0.260,0.262) | 0.252 | (0.251,0.253) | 0.231 | (0.230,0.232) | 0.229 | (0.229,0.230) | 0.234 |
| | 336 | 0.279 | (0.278,0.280) | 0.288 | (0.286,0.289) | 0.272 | (0.270,0.273) | 0.273 | (0.271,0.274) | 0.279 |
| | 720 | 0.342 | (0.341,0.343) | 0.337 | (0.336,0.338) | 0.327 | (0.326,0.328) | 0.327 | (0.326,0.329) | 0.316 |
| Traffic | 96 | 0.267 | (0.267,0.267) | 0.287 | (0.287,0.287) | 0.266 | (0.266,0.266) | 0.269 | (0.269,0.269) | 0.248 |
| | 192 | 0.271 | (0.270,0.271) | 0.290 | (0.290,0.290) | 0.270 | (0.270,0.271) | 0.272 | (0.272,0.272) | 0.247 |
| | 336 | 0.296 | (0.296,0.297) | 0.294 | (0.294,0.294) | 0.278 | (0.278,0.278) | 0.269 | (0.269,0.269) | 0.271 |
| | 720 | 0.291 | (0.291,0.292) | 0.312 | (0.312,0.312) | 0.294 | (0.293,0.294) | 0.294 | (0.294,0.294) | 0.288 |
| Electricity | 96 | 0.233 | (0.233,0.233) | 0.246 | (0.246,0.246) | 0.234 | (0.233,0.234) | 0.229 | (0.229,0.230) | 0.224 |
| | 192 | 0.247 | (0.247,0.247) | 0.257 | (0.257,0.257) | 0.250 | (0.249,0.250) | 0.242 | (0.241,0.242) | 0.241 |
| | 336 | 0.267 | (0.266,0.267) | 0.273 | (0.273,0.274) | 0.263 | (0.263,0.264) | 0.257 | (0.257,0.258) | 0.248 |
| | 720 | 0.290 | (0.290,0.290) | 0.302 | (0.302,0.302) | 0.296 | (0.296,0.296) | 0.290 | (0.290,0.290) | 0.298 |
| **#Wins** | | 5 | | 13 | | 6 | | 8 | | - |

Table 9: Confidence Intervals for MAE predictions of Time-LLM. The best performing model in highlighted in Red color text. #Wins refers to the total number of times the method performed best.

| Models Dataset | Window | Time-LLM MSE | CI | w/o LLM MSE | CI | LLM2Attn MSE | CI | LLM2Trsf MSE | CI | From Paper |
|---|---|---|---|---|---|---|---|---|---|---|
| ETTh1 | 96 | 0.376 | (0.371,0.382) | 0.360 | (0.355,0.364) | 0.377 | (0.371,0.383) | 0.382 | (0.375,0.387) | 0.362 |
| | 192 | 0.407 | (0.401,0.412) | 0.401 | (0.397,0.406) | 0.408 | (0.403,0.414) | 0.416 | (0.411,0.422) | 0.398 |
| | 336 | 0.430 | (0.423,0.437) | 0.431 | (0.425,0.437) | 0.477 | (0.470,0.482) | 0.441 | (0.437,0.446) | 0.430 |
| | 720 | 0.457 | (0.450,0.463) | 0.429 | (0.422,0.434) | 0.429 | (0.423,0.434) | 0.479 | (0.473,0.484) | 0.442 |
| ETTh2 | 96 | 0.286 | (0.282,0.295) | 0.271 | (0.264,0.278) | 0.281 | (0.276,0.288) | 0.292 | (0.284,0.297) | 0.268 |
| | 192 | 0.361 | (0.354,0.367) | 0.342 | (0.336,0.349) | 0.355 | (0.346,0.363) | 0.360 | (0.354,0.367) | 0.329 |
| | 336 | 0.390 | (0.384,0.396) | 0.379 | (0.373,0.384) | 0.384 | (0.377,0.393) | 0.379 | (0.371,0.385) | 0.368 |
| | 720 | 0.405 | (0.397,0.410) | 0.389 | (0.382,0.396) | 0.395 | (0.390,0.400) | 0.405 | (0.397,0.412) | 0.372 |
| ETTm1 | 96 | 0.291 | (0.288,0.293) | 0.292 | (0.289,0.295) | 0.289 | (0.286,0.292) | 0.292 | (0.289,0.295) | 0.272 |
| | 192 | 0.341 | (0.339,0.344) | 0.331 | (0.328,0.334) | 0.336 | (0.334,0.340) | 0.341 | (0.338,0.344) | 0.310 |
| | 336 | 0.359 | (0.356,0.362) | 0.362 | (0.359,0.364) | 0.373 | (0.368,0.376) | 0.374 | (0.370,0.377) | 0.352 |
| | 720 | 0.433 | (0.431,0.436) | 0.417 | (0.414,0.419) | 0.429 | (0.425,0.432) | 0.429 | (0.426,0.433) | 0.383 |
| ETTm2 | 96 | 0.162 | (0.160,0.164) | 0.161 | (0.159,0.163) | 0.163 | (0.160,0.165) | 0.165 | (0.163,0.167) | 0.161 |
| | 192 | 0.235 | (0.232,0.239) | 0.217 | (0.214,0.220) | 0.223 | (0.221,0.227) | 0.222 | (0.219,0.225) | 0.219 |
| | 336 | 0.280 | (0.276,0.282) | 0.272 | (0.268,0.275) | 0.284 | (0.281,0.287) | 0.270 | (0.266,0.273) | 0.271 |
| | 720 | 0.366 | (0.362,0.370) | 0.359 | (0.356,0.362) | 0.367 | (0.363,0.371) | 0.355 | (0.352,0.359) | 0.352 |
| Illness | 24 | 1.792 | (1.651,1.954) | 2.034 | (1.872,2.225) | 1.860 | (1.691,2.059) | 1.923 | (1.755,2.073) | 1.285 |
| | 36 | 1.833 | (1.701,2.009) | 1.923 | (1.753,2.056) | 1.805 | (1.657,1.967) | 1.816 | (1.707,1.961) | 1.404 |
| | 48 | 2.269 | (2.153,2.379) | 1.916 | (1.804,2.024) | 1.716 | (1.601,1.844) | 1.655 | (1.552,1.760) | 1.523 |
| | 60 | 2.177 | (2.064,2.308) | 1.953 | (1.844,2.076) | 1.777 | (1.680,1.874) | 1.789 | (1.637,1.916) | 1.531 |
| Weather | 96 | 0.155 | (0.153,0.157) | 0.171 | (0.168,0.173) | 0.147 | (0.144,0.149) | 0.147 | (0.145,0.150) | 0.147 |
| | 192 | 0.223 | (0.221,0.225) | 0.214 | (0.212,0.217) | 0.191 | (0.189,0.193) | 0.191 | (0.189,0.194) | 0.189 |
| | 336 | 0.251 | (0.249,0.254) | 0.260 | (0.258,0.262) | 0.242 | (0.240,0.245) | 0.246 | (0.243,0.248) | 0.262 |
| | 720 | 0.345 | (0.343,0.348) | 0.328 | (0.325,0.331) | 0.319 | (0.317,0.322) | 0.323 | (0.321,0.326) | 0.304 |
| Traffic | 96 | 0.392 | (0.390,0.394) | 0.409 | (0.407,0.411) | 0.395 | (0.393,0.397) | 0.395 | (0.393,0.397) | 0.362 |
| | 192 | 0.409 | (0.407,0.411) | 0.417 | (0.415,0.420) | 0.405 | (0.403,0.407) | 0.407 | (0.405,0.409) | 0.374 |
| | 336 | 0.434 | (0.432,0.436) | 0.426 | (0.424,0.428) | 0.416 | (0.414,0.418) | 0.412 | (0.410,0.415) | 0.385 |
| | 720 | 0.451 | (0.450,0.454) | 0.461 | (0.459,0.463) | 0.450 | (0.448,0.452) | 0.451 | (0.448,0.453) | 0.430 |
| Electricity | 96 | 0.137 | (0.137,0.138) | 0.143 | (0.143,0.143) | 0.138 | (0.137,0.138) | 0.133 | (0.133,0.134) | 0.131 |
| | 192 | 0.152 | (0.151,0.152) | 0.157 | (0.157,0.158) | 0.154 | (0.154,0.155) | 0.148 | (0.148,0.149) | 0.152 |
| | 336 | 0.169 | (0.169,0.169) | 0.174 | (0.173,0.174) | 0.169 | (0.169,0.170) | 0.164 | (0.164,0.165) | 0.160 |
| | 720 | 0.200 | (0.199,0.200) | 0.211 | (0.210,0.211) | 0.209 | (0.208,0.209) | 0.201 | (0.201,0.202) | 0.192 |
| $1^{st}$ count | | 5 | | 11 | | 10 | | 7 | | - |

Table 10: Confidence Intervals for MSE predictions of Time-LLM. The best performing model in highlighted in Red color text. #Wins refers to the total number of times the method performed best.

| Models | | CALF | | w/o LLM | | LLM2Attn | | LLM2Trsf | | From Paper |
|---|---|---|---|---|---|---|---|---|---|---|
| Dataset | Window | MAE | CI | MAE | CI | MAE | CI | MAE | CI | |
| ETTh1 | 96 | 0.393 | (0.391,0.394) | 0.391 | (0.390,0.392) | 0.391 | (0.389,0.392) | 0.393 | (0.392,0.395) | 0.389 |
| | 192 | 0.426 | (0.424,0.427) | 0.419 | (0.418,0.420) | 0.424 | (0.423,0.425) | 0.423 | (0.422,0.425) | 0.423 |
| | 336 | 0.440 | (0.439,0.441) | 0.437 | (0.436,0.438) | 0.441 | (0.440,0.442) | 0.439 | (0.438,0.440) | 0.436 |
| | 720 | 0.466 | (0.465,0.467) | 0.467 | (0.465,0.467) | 0.467 | (0.466,0.467) | 0.465 | (0.464,0.466) | 0.467 |
| ETTh2 | 96 | 0.336 | (0.333,0.339) | 0.332 | (0.329,0.335) | 0.333 | (0.331,0.336) | 0.333 | (0.331,0.336) | 0.331 |
| | 192 | 0.378 | (0.376,0.381) | 0.378 | (0.376,0.381) | 0.379 | (0.376,0.382) | 0.379 | (0.376,0.382) | 0.380 |
| | 336 | 0.394 | (0.391,0.396) | 0.394 | (0.391,0.398) | 0.393 | (0.390,0.396) | 0.394 | (0.391,0.398) | 0.394 |
| | 720 | 0.428 | (0.425,0.430) | 0.428 | (0.426,0.430) | 0.427 | (0.424,0.430) | 0.427 | (0.425,0.429) | 0.426 |
| ETTm1 | 96 | 0.350 | (0.349,0.352) | 0.348 | (0.346,0.350) | 0.348 | (0.347,0.349) | 0.348 | (0.347,0.349) | 0.349 |
| | 192 | 0.376 | (0.375,0.376) | 0.374 | (0.374,0.375) | 0.376 | (0.375,0.377) | 0.374 | (0.373,0.375) | 0.375 |
| | 336 | 0.401 | (0.400,0.402) | 0.399 | (0.398,0.400) | 0.401 | (0.400,0.401) | 0.399 | (0.398,0.400) | 0.399 |
| | 720 | 0.438 | (0.437,0.439) | 0.442 | (0.441,0.443) | 0.439 | (0.438,0.439) | 0.439 | (0.438,0.440) | 0.438 |
| ETTm2 | 96 | 0.255 | (0.255,0.256) | 0.256 | (0.255,0.257) | 0.253 | (0.252,0.254) | 0.252 | (0.251,0.253) | 0.256 |
| | 192 | 0.300 | (0.299,0.302) | 0.299 | (0.297,0.300) | 0.297 | (0.295,0.298) | 0.297 | (0.296,0.298) | 0.297 |
| | 336 | 0.341 | (0.340,0.343) | 0.340 | (0.338,0.341) | 0.339 | (0.338,0.341) | 0.339 | (0.338,0.341) | 0.339 |
| | 720 | 0.395 | (0.394,0.397) | 0.395 | (0.394,0.397) | 0.395 | (0.394,0.397) | 0.395 | (0.393,0.396) | 0.393 |
| Illness | 24 | 0.788 | (0.742,0.841) | 0.800 | (0.740,0.852) | 0.806 | (0.757,0.845) | 0.792 | (0.745,0.832) | 0.000 |
| | 36 | 0.837 | (0.798,0.872) | 0.802 | (0.771,0.830) | 0.892 | (0.849,0.943) | 0.928 | (0.895,0.974) | 0.000 |
| | 48 | 0.890 | (0.842,0.937) | 0.888 | (0.849,0.933) | 0.897 | (0.868,0.938) | 0.859 | (0.816,0.907) | 0.000 |
| | 60 | 0.962 | (0.917,0.999) | 0.955 | (0.917,0.997) | 0.975 | (0.937,1.022) | 0.861 | (0.832,0.904) | 0.000 |
| Weather | 96 | 0.207 | (0.206,0.207) | 0.212 | (0.212,0.213) | 0.217 | (0.216,0.217) | 0.211 | (0.210,0.211) | 0.204 |
| | 192 | 0.251 | (0.250,0.252) | 0.256 | (0.255,0.257) | 0.257 | (0.256,0.258) | 0.255 | (0.254,0.255) | 0.250 |
| | 336 | 0.292 | (0.291,0.293) | 0.296 | (0.295,0.297) | 0.298 | (0.297,0.299) | 0.296 | (0.295,0.297) | 0.291 |
| | 720 | 0.345 | (0.344,0.347) | 0.347 | (0.346,0.349) | 0.347 | (0.345,0.348) | 0.347 | (0.346,0.349) | 0.352 |
| Traffic | 96 | 0.274 | (0.272,0.275) | 0.267 | (0.266,0.268) | 0.265 | (0.264,0.266) | 0.258 | (0.257,0.260) | 0.268 |
| | 192 | 0.276 | (0.275,0.277) | 0.271 | (0.270,0.273) | 0.268 | (0.267,0.269) | 0.265 | (0.265,0.267) | 0.278 |
| | 336 | 0.286 | (0.285,0.287) | 0.278 | (0.277,0.279) | 0.275 | (0.274,0.276) | 0.272 | (0.271,0.273) | 0.281 |
| | 720 | 0.301 | (0.300,0.302) | 0.297 | (0.296,0.298) | 0.293 | (0.292,0.294) | 0.291 | (0.290,0.291) | 0.300 |
| Electricity | 96 | 0.240 | (0.239,0.240) | 0.238 | (0.238,0.239) | 0.238 | (0.238,0.239) | 0.236 | (0.235,0.236) | 0.238 |
| | 192 | 0.254 | (0.253,0.254) | 0.250 | (0.249,0.251) | 0.251 | (0.250,0.252) | 0.248 | (0.248,0.249) | 0.252 |
| | 336 | 0.270 | (0.270,0.271) | 0.265 | (0.265,0.266) | 0.266 | (0.266,0.267) | 0.264 | (0.263,0.264) | 0.267 |
| | 720 | 0.300 | (0.300,0.301) | 0.297 | (0.296,0.297) | 0.302 | (0.302,0.303) | 0.299 | (0.299,0.300) | 0.303 |
| **#Wins** | | 7 | | 9 | | 4 | | 12 | | - |

Table 11: Confidence Intervals for MAE predictions of CALF. The best performing model in highlighted in Red color text. #Wins refers to the total number of times the method performed best.

| Models | | CALF | | w/o LLM | | LLM2Attn | | LLM2Trsf | | From Paper |
|---|---|---|---|---|---|---|---|---|---|---|
| Dataset | Window | MSE | CI | MSE | CI | MSE | CI | MSE | CI | |
| ETTh1 | 96 | 0.370 | (0.366,0.373) | 0.375 | (0.372,0.378) | 0.368 | (0.364,0.371) | 0.371 | (0.368,0.374) | 0.369 |
| | 192 | 0.429 | (0.426,0.433) | 0.427 | (0.424,0.430) | 0.425 | (0.422,0.428) | 0.428 | (0.424,0.431) | 0.427 |
| | 336 | 0.451 | (0.448,0.454) | 0.457 | (0.454,0.460) | 0.448 | (0.446,0.452) | 0.449 | (0.446,0.451) | 0.456 |
| | 720 | 0.476 | (0.474,0.478) | 0.488 | (0.486,0.491) | 0.471 | (0.470,0.473) | 0.475 | (0.474,0.478) | 0.479 |
| ETTh2 | 96 | 0.284 | (0.279,0.289) | 0.282 | (0.277,0.286) | 0.282 | (0.278,0.286) | 0.282 | (0.278,0.286) | 0.279 |
| | 192 | 0.353 | (0.347,0.359) | 0.354 | (0.349,0.359) | 0.354 | (0.348,0.359) | 0.353 | (0.349,0.358) | 0.353 |
| | 336 | 0.361 | (0.356,0.365) | 0.366 | (0.362,0.371) | 0.360 | (0.356,0.364) | 0.363 | (0.359,0.367) | 0.362 |
| | 720 | 0.406 | (0.403,0.409) | 0.408 | (0.404,0.412) | 0.404 | (0.400,0.408) | 0.404 | (0.400,0.408) | 0.404 |
| ETTm1 | 96 | 0.323 | (0.319,0.329) | 0.323 | (0.320,0.325) | 0.320 | (0.318,0.323) | 0.321 | (0.319,0.324) | 0.323 |
| | 192 | 0.375 | (0.374,0.377) | 0.374 | (0.371,0.376) | 0.375 | (0.373,0.377) | 0.372 | (0.370,0.374) | 0.374 |
| | 336 | 0.411 | (0.409,0.413) | 0.409 | (0.407,0.411) | 0.411 | (0.410,0.414) | 0.408 | (0.406,0.410) | 0.409 |
| | 720 | 0.476 | (0.474,0.478) | 0.484 | (0.482,0.485) | 0.477 | (0.476,0.479) | 0.478 | (0.476,0.480) | 0.477 |
| ETTm2 | 96 | 0.177 | (0.175,0.179) | 0.178 | (0.176,0.180) | 0.176 | (0.174,0.178) | 0.175 | (0.173,0.176) | 0.178 |
| | 192 | 0.245 | (0.243,0.246) | 0.244 | (0.241,0.246) | 0.241 | (0.239,0.243) | 0.242 | (0.240,0.245) | 0.242 |
| | 336 | 0.309 | (0.306,0.311) | 0.308 | (0.305,0.310) | 0.306 | (0.304,0.309) | 0.308 | (0.305,0.310) | 0.307 |
| | 720 | 0.402 | (0.400,0.405) | 0.401 | (0.397,0.404) | 0.401 | (0.398,0.404) | 0.401 | (0.398,0.404) | 0.397 |
| Illness | 24 | 1.460 | (1.297,1.672) | 1.544 | (1.342,1.719) | 1.573 | (1.382,1.758) | 1.450 | (1.267,1.625) | - |
| | 36 | 1.573 | (1.441,1.705) | 1.437 | (1.307,1.533) | 1.699 | (1.540,1.868) | 1.780 | (1.610,1.960) | - |
| | 48 | 1.784 | (1.630,1.937) | 1.710 | (1.500,1.883) | 1.716 | (1.602,1.854) | 1.639 | (1.528,1.821) | - |
| | 60 | 1.982 | (1.786,2.128) | 1.867 | (1.720,2.025) | 2.004 | (1.837,2.186) | 1.652 | (1.522,1.832) | - |
| Weather | 96 | 0.168 | (0.166,0.169) | 0.176 | (0.175,0.177) | 0.178 | (0.177,0.180) | 0.173 | (0.172,0.175) | 0.164 |
| | 192 | 0.216 | (0.214,0.218) | 0.224 | (0.223,0.226) | 0.225 | (0.223,0.226) | 0.221 | (0.220,0.223) | 0.214 |
| | 336 | 0.271 | (0.269,0.273) | 0.277 | (0.275,0.279) | 0.279 | (0.277,0.280) | 0.276 | (0.274,0.278) | 0.269 |
| | 720 | 0.350 | (0.347,0.353) | 0.352 | (0.350,0.355) | 0.352 | (0.349,0.354) | 0.353 | (0.350,0.355) | 0.355 |
| Traffic | 96 | 0.416 | (0.413,0.419) | 0.410 | (0.407,0.413) | 0.402 | (0.399,0.404) | 0.396 | (0.393,0.399) | 0.407 |
| | 192 | 0.430 | (0.429,0.433) | 0.428 | (0.425,0.431) | 0.419 | (0.416,0.421) | 0.416 | (0.414,0.418) | 0.430 |
| | 336 | 0.451 | (0.449,0.453) | 0.443 | (0.441,0.445) | 0.435 | (0.433,0.437) | 0.431 | (0.429,0.432) | 0.444 |
| | 720 | 0.478 | (0.476,0.480) | 0.476 | (0.474,0.479) | 0.467 | (0.465,0.469) | 0.463 | (0.461,0.465) | 0.477 |
| Electricity | 96 | 0.147 | (0.146,0.148) | 0.149 | (0.148,0.151) | 0.148 | (0.146,0.149) | 0.145 | (0.144,0.147) | 0.145 |
| | 192 | 0.163 | (0.162,0.164) | 0.162 | (0.161,0.163) | 0.161 | (0.160,0.162) | 0.159 | (0.158,0.160) | 0.161 |
| | 336 | 0.178 | (0.178,0.179) | 0.175 | (0.175,0.176) | 0.175 | (0.174,0.176) | 0.172 | (0.171,0.173) | 0.175 |
| | 720 | 0.215 | (0.214,0.215) | 0.212 | (0.211,0.213) | 0.217 | (0.217,0.218) | 0.213 | (0.213,0.214) | 0.222 |
| **#Wins** | | 6 | | 4 | | 9 | | 13 | | - |

Table 12: Confidence Intervals for MSE predictions of CALF. The best performing model in highlighted in Red color text. #Wins refers to the total number of times the method performed best. Note that '-' means the dataset has not been included in the original paper.

| Models | | OneFitsAll | | w/o LLM | | LLM2Attn | | LLM2Trsf | | From Paper |
|---|---|---|---|---|---|---|---|---|---|---|
| Dataset | Window | MAE | CI | MAE | CI | MAE | CI | MAE | CI | |
| ETTh1 | 96 | 0.389 | (0.387,0.391) | 0.390 | (0.387,0.392) | 0.406 | (0.404,0.409) | 0.412 | (0.408,0.414) | 0.397 |
| | 192 | 0.413 | (0.411,0.416) | 0.416 | (0.413,0.418) | 0.441 | (0.438,0.443) | 0.455 | (0.452,0.458) | 0.418 |
| | 336 | 0.431 | (0.428,0.433) | 0.430 | (0.428,0.432) | 0.461 | (0.458,0.464) | 0.460 | (0.457,0.462) | 0.433 |
| | 720 | 0.449 | (0.446,0.451) | 0.454 | (0.451,0.457) | 0.501 | (0.498,0.504) | 0.570 | (0.566,0.575) | 0.456 |
| ETTh2 | 96 | 0.335 | (0.333,0.336) | 0.337 | (0.335,0.338) | 0.351 | (0.349,0.353) | 0.348 | (0.346,0.350) | 0.342 |
| | 192 | 0.380 | (0.378,0.381) | 0.382 | (0.380,0.383) | 0.395 | (0.393,0.396) | 0.391 | (0.389,0.392) | 0.389 |
| | 336 | 0.405 | (0.403,0.406) | 0.410 | (0.407,0.411) | 0.416 | (0.414,0.418) | 0.414 | (0.413,0.416) | 0.407 |
| | 720 | 0.436 | (0.435,0.438) | 0.430 | (0.428,0.432) | 0.448 | (0.446,0.450) | 0.439 | (0.437,0.441) | 0.441 |
| ETTm1 | 96 | 0.340 | (0.339,0.340) | 0.342 | (0.341,0.342) | 0.337 | (0.336,0.338) | 0.339 | (0.338,0.340) | 0.346 |
| | 192 | 0.368 | (0.367,0.368) | 0.363 | (0.362,0.363) | 0.366 | (0.366,0.367) | 0.369 | (0.369,0.370) | 0.372 |
| | 336 | 0.386 | (0.386,0.387) | 0.381 | (0.381,0.382) | 0.389 | (0.389,0.390) | 0.387 | (0.386,0.387) | 0.394 |
| | 720 | 0.416 | (0.415,0.416) | 0.413 | (0.412,0.413) | 0.427 | (0.426,0.428) | 0.421 | (0.420,0.422) | 0.421 |
| ETTm2 | 96 | 0.249 | (0.248,0.250) | 0.248 | (0.247,0.248) | 0.251 | (0.251,0.252) | 0.250 | (0.249,0.250) | 0.262 |
| | 192 | 0.291 | (0.291,0.292) | 0.286 | (0.286,0.287) | 0.291 | (0.290,0.292) | 0.288 | (0.287,0.289) | 0.301 |
| | 336 | 0.327 | (0.326,0.328) | 0.323 | (0.322,0.324) | 0.327 | (0.326,0.327) | 0.323 | (0.322,0.324) | 0.341 |
| | 720 | 0.376 | (0.375,0.377) | 0.380 | (0.379,0.381) | 0.382 | (0.381,0.383) | 0.382 | (0.381,0.383) | 0.401 |
| Illness | 24 | 0.823 | (0.806,0.841) | 0.930 | (0.914,0.945) | 0.807 | (0.788,0.824) | 0.846 | (0.830,0.865) | 0.881 |
| | 36 | 0.854 | (0.840,0.868) | 0.913 | (0.901,0.926) | 0.816 | (0.799,0.830) | 0.848 | (0.833,0.864) | 0.892 |
| | 48 | 0.855 | (0.840,0.866) | 0.911 | (0.897,0.923) | 0.846 | (0.828,0.860) | 0.848 | (0.837,0.860) | 0.884 |
| | 60 | 0.877 | (0.867,0.889) | 0.942 | (0.932,0.957) | 0.850 | (0.836,0.863) | 0.861 | (0.850,0.876) | 0.957 |
| Weather | 96 | 0.188 | (0.188,0.189) | 0.212 | (0.212,0.213) | 0.193 | (0.192,0.193) | 0.188 | (0.187,0.188) | 0.212 |
| | 192 | 0.230 | (0.230,0.231) | 0.251 | (0.251,0.252) | 0.231 | (0.230,0.232) | 0.233 | (0.232,0.234) | 0.248 |
| | 336 | 0.273 | (0.272,0.273) | 0.289 | (0.288,0.290) | 0.273 | (0.273,0.274) | 0.275 | (0.274,0.275) | 0.286 |
| | 720 | 0.328 | (0.328,0.329) | 0.339 | (0.339,0.340) | 0.328 | (0.327,0.328) | 0.328 | (0.328,0.329) | 0.337 |
| Traffic | 96 | 0.264 | (0.263,0.264) | 0.264 | (0.264,0.264) | 0.257 | (0.257,0.257) | 0.252 | (0.252,0.252) | 0.282 |
| | 192 | 0.268 | (0.268,0.268) | 0.271 | (0.271,0.271) | 0.260 | (0.260,0.261) | 0.246 | (0.245,0.246) | 0.290 |
| | 336 | 0.273 | (0.273,0.273) | 0.271 | (0.270,0.271) | 0.264 | (0.264,0.265) | 0.255 | (0.254,0.255) | 0.294 |
| | 720 | 0.291 | (0.291,0.291) | 0.289 | (0.289,0.289) | 0.284 | (0.284,0.284) | 0.274 | (0.274,0.274) | 0.312 |
| Electricity | 96 | 0.239 | (0.238,0.239) | 0.230 | (0.229,0.230) | 0.224 | (0.224,0.224) | 0.218 | (0.218,0.218) | 0.238 |
| | 192 | 0.253 | (0.252,0.253) | 0.242 | (0.242,0.242) | 0.238 | (0.238,0.238) | 0.233 | (0.233,0.233) | 0.251 |
| | 336 | 0.266 | (0.266,0.267) | 0.258 | (0.258,0.258) | 0.254 | (0.254,0.254) | 0.250 | (0.250,0.250) | 0.266 |
| | 720 | 0.293 | (0.293,0.294) | 0.290 | (0.290,0.290) | 0.285 | (0.285,0.285) | 0.283 | (0.283,0.283) | 0.297 |
| **#Wins** | | 9 | | 8 | | 6 | | 9 | | - |

Table 13: Confidence Intervals for MAE predictions of OneFitsAll. The best performing model in highlighted in Red color text. #Wins refers to the total number of times the method performed best.

| Models | | OneFitsAll | | w/o LLM | | LLM2Attn | | LLM2Trsf | | From Paper |
|---|---|---|---|---|---|---|---|---|---|---|
| Dataset | Window | MSE | CI | MSE | CI | MSE | CI | MSE | CI | |
| ETTh1 | 96 | 0.370 | (0.365,0.375) | 0.371 | (0.366,0.376) | 0.403 | (0.397,0.409) | 0.397 | (0.391,0.403) | 0.376 |
| | 192 | 0.412 | (0.406,0.417) | 0.416 | (0.410,0.420) | 0.454 | (0.448,0.461) | 0.482 | (0.476,0.489) | 0.416 |
| | 336 | 0.448 | (0.443,0.454) | 0.441 | (0.434,0.446) | 0.483 | (0.475,0.490) | 0.480 | (0.475,0.487) | 0.442 |
| | 720 | 0.441 | (0.436,0.447) | 0.442 | (0.437,0.448) | 0.522 | (0.515,0.527) | 0.743 | (0.732,0.752) | 0.477 |
| ETTh2 | 96 | 0.280 | (0.278,0.283) | 0.284 | (0.282,0.287) | 0.304 | (0.301,0.307) | 0.298 | (0.295,0.301) | 0.285 |
| | 192 | 0.348 | (0.346,0.352) | 0.355 | (0.352,0.357) | 0.370 | (0.367,0.375) | 0.363 | (0.360,0.366) | 0.354 |
| | 336 | 0.380 | (0.377,0.383) | 0.388 | (0.384,0.390) | 0.399 | (0.396,0.401) | 0.392 | (0.389,0.395) | 0.373 |
| | 720 | 0.406 | (0.403,0.409) | 0.400 | (0.398,0.403) | 0.431 | (0.428,0.433) | 0.419 | (0.416,0.422) | 0.406 |
| ETTm1 | 96 | 0.300 | (0.298,0.301) | 0.301 | (0.300,0.303) | 0.299 | (0.297,0.300) | 0.304 | (0.303,0.306) | 0.292 |
| | 192 | 0.343 | (0.342,0.344) | 0.338 | (0.337,0.339) | 0.349 | (0.348,0.351) | 0.356 | (0.355,0.358) | 0.332 |
| | 336 | 0.376 | (0.374,0.377) | 0.369 | (0.368,0.370) | 0.383 | (0.382,0.385) | 0.379 | (0.378,0.381) | 0.366 |
| | 720 | 0.431 | (0.430,0.433) | 0.425 | (0.424,0.427) | 0.447 | (0.446,0.449) | 0.438 | (0.437,0.440) | 0.417 |
| ETTm2 | 96 | 0.163 | (0.162,0.164) | 0.163 | (0.162,0.164) | 0.165 | (0.164,0.166) | 0.164 | (0.163,0.165) | 0.173 |
| | 192 | 0.222 | (0.220,0.223) | 0.221 | (0.219,0.222) | 0.223 | (0.221,0.224) | 0.219 | (0.218,0.220) | 0.229 |
| | 336 | 0.273 | (0.272,0.275) | 0.275 | (0.273,0.276) | 0.275 | (0.273,0.276) | 0.272 | (0.270,0.273) | 0.286 |
| | 720 | 0.357 | (0.355,0.359) | 0.364 | (0.363,0.366) | 0.362 | (0.360,0.364) | 0.362 | (0.360,0.363) | 0.378 |
| Illness | 24 | 1.869 | (1.780,1.950) | 2.119 | (2.022,2.215) | 1.799 | (1.711,1.887) | 1.929 | (1.868,2.020) | 2.063 |
| | 36 | 1.853 | (1.791,1.925) | 1.929 | (1.857,1.997) | 1.727 | (1.660,1.800) | 1.801 | (1.741,1.859) | 1.868 |
| | 48 | 1.886 | (1.828,1.942) | 1.883 | (1.827,1.932) | 1.804 | (1.744,1.866) | 1.807 | (1.749,1.869) | 1.790 |
| | 60 | 1.877 | (1.827,1.937) | 1.911 | (1.861,1.953) | 1.724 | (1.672,1.776) | 1.784 | (1.736,1.841) | 1.979 |
| Weather | 96 | 0.148 | (0.147,0.150) | 0.173 | (0.172,0.175) | 0.150 | (0.149,0.152) | 0.149 | (0.148,0.150) | 0.162 |
| | 192 | 0.192 | (0.191,0.194) | 0.216 | (0.215,0.218) | 0.192 | (0.191,0.194) | 0.196 | (0.194,0.197) | 0.204 |
| | 336 | 0.246 | (0.244,0.247) | 0.263 | (0.261,0.264) | 0.244 | (0.243,0.246) | 0.247 | (0.246,0.248) | 0.254 |
| | 720 | 0.320 | (0.318,0.321) | 0.330 | (0.329,0.332) | 0.318 | (0.316,0.319) | 0.322 | (0.321,0.324) | 0.326 |
| Traffic | 96 | 0.396 | (0.393,0.398) | 0.422 | (0.419,0.424) | 0.393 | (0.391,0.395) | 0.391 | (0.389,0.393) | 0.388 |
| | 192 | 0.412 | (0.410,0.414) | 0.430 | (0.428,0.432) | 0.406 | (0.404,0.408) | 0.395 | (0.393,0.397) | 0.407 |
| | 336 | 0.421 | (0.419,0.423) | 0.437 | (0.435,0.439) | 0.413 | (0.412,0.416) | 0.406 | (0.405,0.408) | 0.412 |
| | 720 | 0.455 | (0.453,0.457) | 0.470 | (0.468,0.472) | 0.451 | (0.449,0.453) | 0.444 | (0.442,0.446) | 0.450 |
| Electricity | 96 | 0.141 | (0.141,0.142) | 0.137 | (0.136,0.137) | 0.133 | (0.133,0.134) | 0.128 | (0.128,0.129) | 0.139 |
| | 192 | 0.158 | (0.157,0.158) | 0.151 | (0.151,0.152) | 0.149 | (0.149,0.150) | 0.145 | (0.145,0.146) | 0.153 |
| | 336 | 0.172 | (0.172,0.172) | 0.167 | (0.167,0.168) | 0.165 | (0.164,0.165) | 0.160 | (0.160,0.161) | 0.169 |
| | 720 | 0.207 | (0.207,0.208) | 0.206 | (0.205,0.206) | 0.201 | (0.201,0.202) | 0.196 | (0.195,0.196) | 0.206 |
| **#Wins** | | 10 | | 5 | | 7 | | 10 | | - |

Table 14: Confidence Intervals for MSE predictions of OneFitsAll. The best performing model in highlighted in Red color text. #Wins refers to the total number of times the method performed best.

| Models | | Time-LLM | | w/o LLM | | LLM2Attn | | LLM2Trsf | |
|---|---|---|---|---|---|---|---|---|---|
| Dataset | Window | MAE | CI | MAE | CI | MAE | CI | MAE | CI |
| Exchange Rate | 96 | 0.251 | (0.248,0.255) | 0.209 | (0.206,0.212) | 0.213 | (0.210,0.217) | 0.239 | (0.237,0.242) |
| | 192 | 0.344 | (0.340,0.349) | 0.305 | (0.301,0.309) | 0.330 | (0.325,0.334) | 0.331 | (0.326,0.336) |
| | 336 | 0.451 | (0.446,0.456) | 0.423 | (0.417,0.428) | 0.471 | (0.465,0.477) | 0.453 | (0.447,0.458) |
| | 720 | 0.771 | (0.761,0.782) | 0.719 | (0.709,0.727) | 0.762 | (0.752,0.771) | 0.762 | (0.753,0.771) |
| Covid Deaths | 30 | 0.090 | (0.018,0.193) | 0.080 | (0.015,0.178) | 0.059 | (0.007,0.125) | 0.055 | (0.008,0.113) |
| Taxi (30 Min) | 48 | 0.275 | (0.270,0.280) | 0.286 | (0.280,0.292) | 0.269 | (0.265,0.275) | 0.256 | (0.251,0.261) |
| NN5 (Daily) | 56 | 0.432 | (0.358,0.500) | 0.425 | (0.360,0.498) | 0.411 | (0.341,0.484) | 0.401 | (0.327,0.465) |
| FRED-MD | 12 | 6.0e-04 | (5.2e-04,7.9e-04) | 2.0e-04 | (1.4e-04,2.7e-04) | 4.7e-03 | (4.5e-03,4.8e-03) | 9.0e-04 | (6.7e-04,1.1e-03) |
| **#Wins** | | 0 | | 5 | | 0 | | 3 | |

| Models | | Time-LLM | | w/o LLM | | LLM2Attn | | LLM2Trsf | |
|---|---|---|---|---|---|---|---|---|---|
| Dataset | Window | MSE | CI | MSE | CI | MSE | CI | MSE | CI |
| Exchange Rate | 96 | 0.123 | (0.120,0.127) | 0.090 | (0.087,0.093) | 0.090 | (0.087,0.093) | 0.110 | (0.107,0.113) |
| | 192 | 0.224 | (0.217,0.230) | 0.185 | (0.180,0.190) | 0.211 | (0.206,0.217) | 0.211 | (0.206,0.216) |
| | 336 | 0.377 | (0.369,0.387) | 0.341 | (0.332,0.348) | 0.407 | (0.398,0.418) | 0.384 | (0.376,0.392) |
| | 720 | 1.018 | (0.997,1.041) | 0.922 | (0.896,0.943) | 1.022 | (1.007,1.043) | 0.996 | (0.975,1.023) |
| Covid Deaths | 30 | 0.194 | (0.000,0.467) | 0.199 | (0.004,0.462) | 0.087 | (0.009,0.196) | 0.082 | (0.002,0.183) |
| Taxi (30 Min) | 48 | 0.161 | (0.154,0.169) | 0.177 | (0.168,0.188) | 0.157 | (0.149,0.164) | 0.141 | (0.134,0.148) |
| NN5 (Daily) | 56 | 0.404 | (0.273,0.548) | 0.379 | (0.271,0.525) | 0.365 | (0.233,0.539) | 0.347 | (0.234,0.474) |
| FRED-MD | 12 | 1.4e-06 | (7.5e-07,2.2e-06) | 2.8e-07 | (8.0e-08,5.6e-07) | 2.5e-05 | (2.4e-05,2.7e-05) | 2.6e-06 | (1.7e-06,3.7e-06) |
| **#Wins** | | 0 | | 5 | | 0 | | 3 | |

Table 15: Confidence Intervals for MAE and MSE predictions of Time-LLM. The best performing model in highlighted in Red color text. #Wins refers to the total number of times the method performed best. The ablation results in the table above are from datasets that have not been studied by the reference methods [50, 15, 22].

| Models | | CALF | | w/o LLM | | LLM2Attn | | LLM2Trsf | |
|---|---|---|---|---|---|---|---|---|---|
| Dataset | Window | MAE | CI | MAE | CI | MAE | CI | MAE | CI |
| Exchange Rate | 96 | 0.203 | (0.200,0.206) | 0.207 | (0.204,0.211) | 0.206 | (0.203,0.209) | 0.207 | (0.204,0.210) |
| | 192 | 0.306 | (0.300,0.312) | 0.305 | (0.300,0.311) | 0.305 | (0.298,0.309) | 0.305 | (0.298,0.309) |
| | 336 | 0.427 | (0.420,0.435) | 0.426 | (0.419,0.436) | 0.427 | (0.418,0.435) | 0.415 | (0.406,0.424) |
| | 720 | 0.732 | (0.721,0.745) | 0.697 | (0.687,0.710) | 0.731 | (0.716,0.750) | 0.708 | (0.695,0.723) |
| Covid Deaths | 30 | 0.084 | (0.025,0.174) | 0.066 | (0.012,0.144) | 0.131 | (0.027,0.274) | 0.066 | (0.014,0.151) |
| Taxi (30 Min) | 48 | 0.258 | (0.254,0.263) | 0.264 | (0.259,0.268) | 0.267 | (0.262,0.271) | 0.267 | (0.263,0.272) |
| NN5 (Daily) | 56 | 0.403 | (0.348,0.471) | 0.386 | (0.325,0.432) | 0.433 | (0.367,0.504) | 0.431 | (0.357,0.495) |
| FRED-MD | 12 | 1.3e-03 | (1.2e-03,1.4e-03) | 1.2e-03 | (1.0e-03,1.3e-03) | 1.6e-03 | (1.4e-03,1.7e-03) | 1.7e-03 | (1.6e-03,1.9e-03) |
| **#Wins** | | 2 | | 4 | | 1 | | 1 | |

| Models | | CALF | | w/o LLM | | LLM2Attn | | LLM2Trsf | |
|---|---|---|---|---|---|---|---|---|---|
| Dataset | Window | MSE | CI | MSE | CI | MSE | CI | MSE | CI |
| Exchange Rate | 96 | 0.083 | (0.080,0.085) | 0.086 | (0.084,0.089) | 0.086 | (0.084,0.089) | 0.087 | (0.085,0.090) |
| | 192 | 0.186 | (0.180,0.193) | 0.183 | (0.177,0.188) | 0.182 | (0.177,0.189) | 0.181 | (0.175,0.186) |
| | 336 | 0.350 | (0.339,0.361) | 0.345 | (0.335,0.360) | 0.345 | (0.332,0.356) | 0.324 | (0.310,0.337) |
| | 720 | 0.935 | (0.906,0.965) | 0.854 | (0.833,0.877) | 0.943 | (0.911,0.978) | 0.878 | (0.856,0.900) |
| Covid Deaths | 30 | 0.163 | (0.005,0.373) | 0.115 | (0.003,0.287) | 0.431 | (0.010,1.046) | 0.106 | (0.008,0.207) |
| Taxi (30 Min) | 48 | 0.142 | (0.134,0.152) | 0.147 | (0.140,0.158) | 0.150 | (0.143,0.157) | 0.150 | (0.142,0.158) |
| NN5 (Daily) | 56 | 0.362 | (0.275,0.522) | 0.336 | (0.236,0.451) | 0.405 | (0.277,0.618) | 0.399 | (0.272,0.585) |
| FRED-MD | 12 | 2.9e-06 | (2.0e-06,4.5e-06) | 2.7e-06 | (2.0e-06,3.3e-06) | 4.9e-06 | (3.4e-06,6.8e-06) | 4.5e-06 | (3.7e-06,5.4e-06) |
| **#Wins** | | 2 | | 3 | | 0 | | 3 | |

Table 16: Confidence Intervals for MAE and MSE predictions of CALF. The best performing model in highlighted in Red color text. #Wins refers to the total number of times the method performed best. The ablation results in the table above are from datasets that have not been studied by the reference methods [50, 15, 22].

| Models | | OneFitsAll | | w/o LLM | | LLM2Attn | | LLM2Trsf | |
|---|---|---|---|---|---|---|---|---|---|
| Dataset | Window | MAE | CI | MAE | CI | MAE | CI | MAE | CI |
| Exchange Rate | 96 | 0.218 | (0.215,0.221) | 0.204 | (0.202,0.207) | 0.215 | (0.213,0.218) | 0.224 | (0.221,0.226) |
| | 192 | 0.307 | (0.303,0.311) | 0.308 | (0.304,0.312) | 0.339 | (0.334,0.344) | 0.343 | (0.340,0.347) |
| | 336 | 0.461 | (0.455,0.467) | 0.452 | (0.446,0.457) | 0.469 | (0.463,0.477) | 0.434 | (0.430,0.440) |
| | 720 | 0.767 | (0.756,0.777) | 0.727 | (0.716,0.736) | 0.735 | (0.722,0.747) | 0.781 | (0.771,0.790) |
| Covid Deaths | 30 | 0.057 | (0.019,0.122) | 0.050 | (0.004,0.100) | 0.058 | (0.019,0.114) | 0.078 | (0.015,0.175) |
| Taxi (30 Min) | 48 | 0.260 | (0.256,0.267) | 0.265 | (0.259,0.271) | 0.264 | (0.258,0.272) | 0.263 | (0.256,0.268) |
| NN5 (Daily) | 56 | 0.438 | (0.364,0.517) | 0.422 | (0.352,0.497) | 0.423 | (0.363,0.480) | 0.420 | (0.354,0.487) |
| FRED-MD | 12 | 6.7e-04 | (5.6e-04,7.7e-04) | 2.5e-04 | (1.9e-04,3.3e-04) | 6.1e-04 | (4.9e-04,7.1e-04) | 1.2e-03 | (1.2e-03,1.3e-03) |
| **#Wins** | | 2 | | 4 | | 0 | | 2 | |

| Models | | OneFitsAll | | w/o LLM | | LLM2Attn | | LLM2Trsf | |
|---|---|---|---|---|---|---|---|---|---|
| Dataset | Window | MSE | CI | MSE | CI | MSE | CI | MSE | CI |
| Exchange Rate | 96 | 0.096 | (0.093,0.098) | 0.086 | (0.084,0.089) | 0.092 | (0.090,0.095) | 0.103 | (0.100,0.106) |
| | 192 | 0.182 | (0.177,0.186) | 0.189 | (0.184,0.194) | 0.232 | (0.224,0.239) | 0.228 | (0.221,0.233) |
| | 336 | 0.402 | (0.393,0.414) | 0.392 | (0.382,0.401) | 0.462 | (0.444,0.481) | 0.359 | (0.351,0.368) |
| | 720 | 1.055 | (1.028,1.084) | 0.932 | (0.909,0.951) | 0.985 | (0.958,1.010) | 1.113 | (1.090,1.146) |
| Covid Deaths | 30 | 0.075 | (0.004,0.161) | 0.073 | (0.002,0.175) | 0.103 | (0.003,0.265) | 0.162 | (0.009,0.435) |
| Taxi (30 Min) | 48 | 0.148 | (0.139,0.158) | 0.150 | (0.141,0.159) | 0.150 | (0.143,0.159) | 0.149 | (0.141,0.158) |
| NN5 (Daily) | 56 | 0.438 | (0.309,0.555) | 0.385 | (0.252,0.508) | 0.390 | (0.262,0.542) | 0.385 | (0.254,0.515) |
| FRED-MD | 12 | 1.2e-06 | (7.4e-07,1.9e-06) | 3.8e-07 | (1.0e-07,7.6e-07) | 1.5e-06 | (8.4e-07,2.2e-06) | 2.4e-06 | (2.1e-06,3.0e-06) |
| **#Wins** | | 2 | | 5 | | 0 | | 1 | |

Table 17: Confidence Intervals for MAE and MSE predictions of OneFitsAll. The best performing model in highlighted in Red color text. #Wins refers to the total number of times the method performed best.

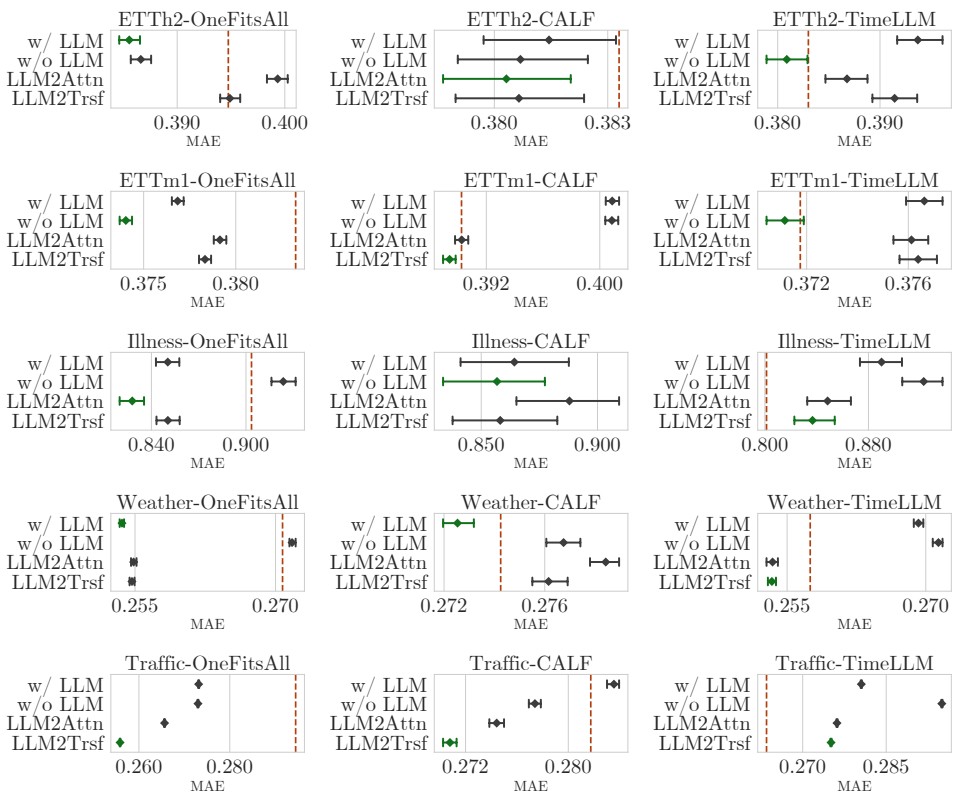

Figure 5: Ablation studies indicate that when different methods remove the LLM ("**w/o LLM**") or replace it with a single-layer attention ("**LLM2Attn**") or Transformer ("**LLM2Trsf**"), the performance on time series forecasting tasks with MAE metric does not decline and even improves, compared with original methods, such as "GPT-2" or "LLaMA". The vertical dashed line in the figures represents the results from the original paper. Above figures are from 'ETTh2', 'ETTm1', 'Illness', 'Weather', and 'Traffic' datasets.

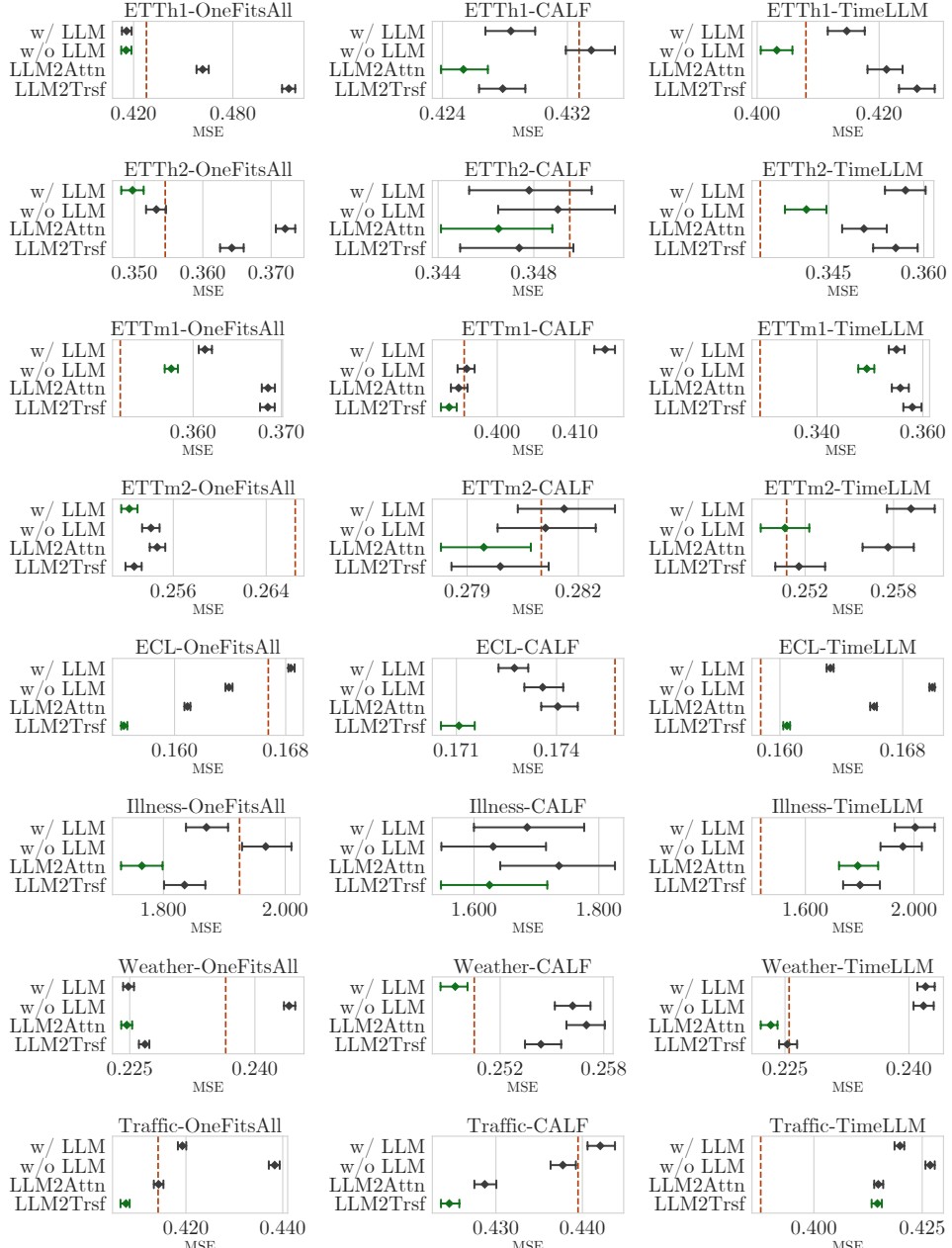

Figure 6: Ablation studies indicate that when different methods remove the LLM ("**w/o LLM**") or replace it with a single-layer attention ("**LLM2Attn**") or Transformer ("**LLM2Trsf**"), the performance on time series forecasting tasks with MSE metric does not decline and even improves, compared with original methods, such as "GPT-2" or "LLaMA". The vertical dashed line in the figures represents the results from the original paper.

| Methods | | CALF [22] | | OneFitAll [50] | | Time-LLM [15] | | D-LTrsf | | LTrsf | | PAttn | | D-PAttn | | DLinear | | MeanP | | Seasonal | |
|---|---|---|---|---|---|---|---|---|---|---|---|---|---|---|---|---|---|---|---|---|---|
| Metric | | MAE | MSE | MAE | MSE | MAE | MSE | MAE | MSE | MAE | MSE | MAE | MSE | MAE | MSE | MAE | MSE | MAE | MSE | MAE | MSE |
| ETTh1 | 96 | 0.389 | 0.369 | 0.397 | 0.376 | 0.392 | 0.362 | 0.392 | 0.379 | 0.390 | 0.371 | 0.387 | 0.383 | 0.387 | 0.376 | 0.399 | 0.375 | 0.525 | 0.753 | 0.799 | 1.185 |
| | 192 | 0.423 | 0.427 | 0.418 | 0.416 | 0.418 | 0.398 | 0.421 | 0.436 | 0.418 | 0.428 | 0.416 | 0.439 | 0.415 | 0.430 | 0.416 | 0.405 | 0.544 | 0.775 | 0.788 | 1.130 |
| | 336 | 0.436 | 0.456 | 0.433 | 0.442 | 0.427 | 0.430 | 0.427 | 0.447 | 0.426 | 0.449 | 0.419 | 0.447 | 0.419 | 0.440 | 0.443 | 0.439 | 0.548 | 0.762 | 0.785 | 1.099 |
| | 720 | 0.467 | 0.479 | 0.456 | 0.477 | 0.457 | 0.442 | 0.460 | 0.470 | 0.465 | 0.485 | 0.457 | 0.474 | 0.455 | 0.469 | 0.490 | 0.472 | 0.549 | 0.720 | 0.789 | 1.083 |
| ETTh2 | 96 | 0.331 | 0.279 | 0.342 | 0.285 | 0.328 | 0.268 | 0.332 | 0.281 | 0.333 | 0.285 | 0.327 | 0.277 | 0.327 | 0.276 | 0.353 | 0.289 | 0.378 | 0.368 | 1.231 | 2.777 |
| | 192 | 0.380 | 0.353 | 0.389 | 0.354 | 0.375 | 0.329 | 0.379 | 0.353 | 0.376 | 0.351 | 0.373 | 0.345 | 0.373 | 0.347 | 0.418 | 0.383 | 0.422 | 0.444 | 1.294 | 2.987 |
| | 336 | 0.394 | 0.362 | 0.407 | 0.373 | 0.409 | 0.368 | 0.382 | 0.350 | 0.379 | 0.346 | 0.375 | 0.338 | 0.376 | 0.341 | 0.465 | 0.448 | 0.450 | 0.476 | 1.313 | 3.049 |
| | 720 | 0.426 | 0.404 | 0.441 | 0.406 | 0.420 | 0.372 | 0.431 | 0.411 | 0.426 | 0.406 | 0.425 | 0.403 | 0.429 | 0.408 | 0.551 | 0.605 | 0.456 | 0.469 | 1.324 | 3.073 |
| ETTm1 | 96 | 0.349 | 0.323 | 0.346 | 0.292 | 0.334 | 0.272 | 0.347 | 0.300 | 0.351 | 0.300 | 0.342 | 0.301 | 0.346 | 0.300 | 0.343 | 0.299 | 0.512 | 0.738 | 0.752 | 1.056 |
| | 192 | 0.375 | 0.374 | 0.372 | 0.332 | 0.358 | 0.310 | 0.367 | 0.335 | 0.366 | 0.327 | 0.366 | 0.344 | 0.370 | 0.343 | 0.365 | 0.335 | 0.518 | 0.745 | 0.769 | 1.075 |
| | 336 | 0.399 | 0.409 | 0.394 | 0.366 | 0.384 | 0.352 | 0.383 | 0.359 | 0.386 | 0.356 | 0.383 | 0.369 | 0.388 | 0.370 | 0.386 | 0.369 | 0.524 | 0.752 | 0.778 | 1.086 |
| | 720 | 0.438 | 0.477 | 0.421 | 0.417 | 0.411 | 0.383 | 0.414 | 0.418 | 0.416 | 0.421 | 0.413 | 0.426 | 0.418 | 0.431 | 0.421 | 0.425 | 0.538 | 0.764 | 0.782 | 1.089 |
| ETTm2 | 96 | 0.256 | 0.178 | 0.262 | 0.173 | 0.253 | 0.161 | 0.256 | 0.168 | 0.258 | 0.172 | 0.249 | 0.164 | 0.252 | 0.165 | 0.260 | 0.167 | 0.345 | 0.312 | 1.208 | 2.730 |
| | 192 | 0.297 | 0.242 | 0.301 | 0.229 | 0.293 | 0.219 | 0.294 | 0.225 | 0.295 | 0.226 | 0.292 | 0.224 | 0.291 | 0.220 | 0.303 | 0.224 | 0.358 | 0.335 | 1.285 | 2.943 |
| | 336 | 0.339 | 0.307 | 0.341 | 0.286 | 0.392 | 0.271 | 0.322 | 0.268 | 0.323 | 0.269 | 0.319 | 0.266 | 0.319 | 0.263 | 0.343 | 0.281 | 0.376 | 0.364 | 1.318 | 3.036 |
| | 720 | 0.393 | 0.397 | 0.401 | 0.378 | 0.379 | 0.352 | 0.372 | 0.344 | 0.372 | 0.344 | 0.370 | 0.345 | 0.369 | 0.339 | 0.421 | 0.397 | 0.418 | 0.433 | 1.339 | 3.109 |
| **1st and 2nd Wins** | | **11** | | **33** | | **0** | | **3** | | **2** | | **34** | | **21** | | **3** | | **0** | | **0** | |
| Illness | 24 | - | - | 0.881 | 2.063 | 0.727 | 1.285 | 0.814 | 1.602 | 0.826 | 1.706 | 0.800 | 1.711 | 0.817 | 1.685 | 1.081 | 2.215 | 1.393 | 4.453 | 1.854 | 7.099 |
| | 36 | - | - | 0.892 | 1.868 | 0.814 | 1.404 | 0.895 | 1.738 | 0.927 | 2.300 | 0.794 | 1.636 | 0.853 | 1.739 | 0.963 | 1.963 | 1.410 | 4.524 | 1.897 | 7.231 |
| | 48 | - | - | 0.884 | 1.790 | 0.807 | 1.523 | 0.931 | 1.897 | 0.992 | 2.534 | 0.853 | 1.767 | 0.874 | 1.775 | 1.024 | 2.130 | 1.444 | 4.819 | 1.882 | 7.205 |
| | 60 | - | - | 0.957 | 1.979 | 0.857 | 1.531 | 0.979 | 2.049 | 0.992 | 2.490 | 0.829 | 1.658 | 0.853 | 1.695 | 1.096 | 2.368 | 1.492 | 5.058 | 1.892 | 7.096 |
| Weather | 96 | 0.204 | 0.164 | 0.212 | 0.162 | 0.201 | 0.147 | 0.203 | 0.160 | 0.204 | 0.161 | 0.191 | 0.151 | 0.195 | 0.152 | 0.237 | 0.176 | 0.312 | 0.297 | 0.550 | 0.576 |
| | 192 | 0.250 | 0.214 | 0.248 | 0.204 | 0.234 | 0.189 | 0.242 | 0.202 | 0.246 | 0.205 | 0.232 | 0.193 | 0.236 | 0.196 | 0.282 | 0.220 | 0.323 | 0.316 | 0.577 | 0.605 |
| | 336 | 0.291 | 0.269 | 0.286 | 0.254 | 0.279 | 0.262 | 0.283 | 0.253 | 0.283 | 0.253 | 0.273 | 0.244 | 0.277 | 0.247 | 0.319 | 0.265 | 0.341 | 0.343 | 0.594 | 0.625 |
| | 720 | 0.352 | 0.355 | 0.337 | 0.326 | 0.316 | 0.304 | 0.333 | 0.320 | 0.332 | 0.322 | 0.323 | 0.311 | 0.330 | 0.316 | 0.362 | 0.323 | 0.369 | 0.384 | 0.605 | 0.633 |
| Traffic | 96 | 0.268 | 0.407 | 0.282 | 0.388 | 0.248 | 0.362 | 0.264 | 0.383 | 0.234 | 0.357 | 0.262 | 0.394 | 0.262 | 0.394 | 0.282 | 0.410 | 0.771 | 1.441 | 0.884 | 1.765 |
| | 192 | 0.278 | 0.430 | 0.290 | 0.407 | 0.247 | 0.374 | 0.275 | 0.399 | 0.243 | 0.377 | 0.265 | 0.404 | 0.266 | 0.404 | 0.287 | 0.423 | 0.774 | 1.455 | 0.844 | 1.628 |
| | 336 | 0.281 | 0.444 | 0.294 | 0.412 | 0.271 | 0.385 | 0.284 | 0.411 | 0.250 | 0.388 | 0.270 | 0.413 | 0.269 | 0.412 | 0.296 | 0.436 | 0.776 | 1.469 | 0.827 | 1.574 |
| | 720 | 0.300 | 0.477 | 0.312 | 0.450 | 0.288 | 0.430 | 0.309 | 0.453 | 0.273 | 0.428 | 0.285 | 0.448 | 0.289 | 0.449 | 0.315 | 0.466 | 0.780 | 1.488 | 0.816 | 1.543 |
| Electricity | 96 | 0.238 | 0.145 | 0.238 | 0.139 | 0.224 | 0.131 | 0.226 | 0.132 | 0.220 | 0.130 | 0.226 | 0.134 | 0.227 | 0.134 | 0.237 | 0.140 | 0.738 | 0.944 | 0.891 | 1.201 |
| | 192 | 0.252 | 0.161 | 0.251 | 0.153 | 0.241 | 0.152 | 0.247 | 0.154 | 0.239 | 0.149 | 0.239 | 0.149 | 0.241 | 0.149 | 0.249 | 0.153 | 0.742 | 0.953 | 0.861 | 1.107 |
| | 336 | 0.267 | 0.175 | 0.266 | 0.169 | 0.248 | 0.160 | 0.265 | 0.171 | 0.254 | 0.164 | 0.257 | 0.166 | 0.257 | 0.166 | 0.267 | 0.169 | 0.747 | 0.966 | 0.847 | 1.064 |
| | 720 | 0.303 | 0.222 | 0.297 | 0.206 | 0.298 | 0.192 | 0.299 | 0.211 | 0.278 | 0.193 | 0.288 | 0.204 | 0.289 | 0.204 | 0.301 | 0.203 | 0.762 | 0.998 | 0.838 | 1.031 |
| **1st and 2nd Wins** | | | | | | **0** | | **0** | | **16** | | **4** | | **1** | | **0** | | **0** | | **0** | |

Table 18: Comparision between simple methods and original results from reference paper [22, 50, 15]. Red: Best performance. Blue: Second Best.

| | | Simple Methods | | | | LLM-Based Methods | | | | | | Other State-of-the-art Methods | | | | | | | | | | | |
|---|---|---|---|---|---|---|---|---|---|---|---|---|---|---|---|---|---|---|---|---|---|---|---|
| Methods | | LTrsf | | PAttn | | CALF [22] | | OneFitsAll [50] | | Time-LLM [15] | | DLinear [46] | | RLinear [21] | | PatchTST [27] | | iTransformer [24] | | TimesNet [41] | | FEDformer [49] | |
| Metric | | MAE | MSE | MAE | MSE | MAE | MSE | MAE | MSE | MAE | MSE | MAE | MSE | MAE | MSE | MAE | MSE | MAE | MSE | MAE | MSE | MAE | MSE |
| ETTm1 | 96 | 0.351 | 0.300 | 0.342 | 0.301 | 0.349 | 0.323 | 0.346 | 0.292 | 0.334 | 0.272 | 0.343 | 0.299 | 0.376 | 0.355 | 0.342 | 0.290 | 0.368 | 0.334 | 0.375 | 0.338 | 0.419 | 0.379 |
| | 192 | 0.366 | 0.327 | 0.366 | 0.344 | 0.375 | 0.374 | 0.372 | 0.332 | 0.358 | 0.310 | 0.365 | 0.335 | 0.392 | 0.391 | 0.369 | 0.332 | 0.391 | 0.377 | 0.387 | 0.374 | 0.441 | 0.426 |
| | 336 | 0.386 | 0.356 | 0.383 | 0.369 | 0.399 | 0.409 | 0.394 | 0.366 | 0.384 | 0.352 | 0.386 | 0.369 | 0.415 | 0.424 | 0.392 | 0.366 | 0.420 | 0.426 | 0.411 | 0.410 | 0.459 | 0.445 |
| | 720 | 0.416 | 0.421 | 0.413 | 0.426 | 0.438 | 0.477 | 0.421 | 0.417 | 0.411 | 0.383 | 0.421 | 0.425 | 0.450 | 0.487 | 0.420 | 0.416 | 0.459 | 0.491 | 0.450 | 0.478 | 0.490 | 0.543 |
| ETTm2 | 96 | 0.258 | 0.172 | 0.249 | 0.164 | 0.256 | 0.178 | 0.262 | 0.173 | 0.253 | 0.161 | 0.260 | 0.167 | 0.265 | 0.182 | 0.255 | 0.165 | 0.264 | 0.180 | 0.267 | 0.187 | 0.287 | 0.203 |
| | 192 | 0.295 | 0.226 | 0.292 | 0.224 | 0.297 | 0.242 | 0.301 | 0.229 | 0.293 | 0.219 | 0.303 | 0.224 | 0.304 | 0.246 | 0.292 | 0.220 | 0.309 | 0.250 | 0.309 | 0.249 | 0.328 | 0.269 |
| | 336 | 0.323 | 0.269 | 0.319 | 0.266 | 0.339 | 0.307 | 0.341 | 0.286 | 0.392 | 0.271 | 0.343 | 0.281 | 0.342 | 0.307 | 0.329 | 0.274 | 0.348 | 0.311 | 0.351 | 0.321 | 0.366 | 0.325 |
| | 720 | 0.372 | 0.344 | 0.370 | 0.345 | 0.393 | 0.397 | 0.401 | 0.378 | 0.379 | 0.352 | 0.421 | 0.397 | 0.398 | 0.407 | 0.385 | 0.362 | 0.407 | 0.412 | 0.403 | 0.408 | 0.415 | 0.421 |
| ETTh1 | 96 | 0.390 | 0.371 | 0.387 | 0.383 | 0.389 | 0.369 | 0.397 | 0.376 | 0.392 | 0.362 | 0.399 | 0.375 | 0.395 | 0.386 | 0.399 | 0.370 | 0.405 | 0.386 | 0.402 | 0.384 | 0.419 | 0.376 |
| | 192 | 0.418 | 0.428 | 0.416 | 0.439 | 0.423 | 0.427 | 0.418 | 0.416 | 0.418 | 0.398 | 0.416 | 0.405 | 0.424 | 0.437 | 0.421 | 0.413 | 0.436 | 0.441 | 0.429 | 0.436 | 0.448 | 0.420 |
| | 336 | 0.426 | 0.449 | 0.419 | 0.447 | 0.436 | 0.456 | 0.433 | 0.442 | 0.427 | 0.430 | 0.443 | 0.439 | 0.446 | 0.479 | 0.436 | 0.422 | 0.458 | 0.487 | 0.469 | 0.491 | 0.465 | 0.459 |
| | 720 | 0.465 | 0.485 | 0.457 | 0.474 | 0.467 | 0.479 | 0.456 | 0.477 | 0.457 | 0.442 | 0.490 | 0.472 | 0.470 | 0.481 | 0.466 | 0.447 | 0.491 | 0.503 | 0.500 | 0.521 | 0.507 | 0.506 |
| ETTh2 | 96 | 0.333 | 0.285 | 0.327 | 0.277 | 0.331 | 0.279 | 0.342 | 0.285 | 0.328 | 0.268 | 0.353 | 0.289 | 0.338 | 0.288 | 0.336 | 0.274 | 0.349 | 0.297 | 0.374 | 0.340 | 0.397 | 0.358 |
| | 192 | 0.376 | 0.351 | 0.373 | 0.345 | 0.380 | 0.353 | 0.389 | 0.354 | 0.375 | 0.329 | 0.418 | 0.383 | 0.390 | 0.374 | 0.379 | 0.339 | 0.400 | 0.380 | 0.414 | 0.402 | 0.439 | 0.429 |
| | 336 | 0.379 | 0.346 | 0.375 | 0.338 | 0.394 | 0.362 | 0.407 | 0.373 | 0.409 | 0.368 | 0.465 | 0.448 | 0.426 | 0.415 | 0.380 | 0.329 | 0.432 | 0.428 | 0.452 | 0.452 | 0.487 | 0.496 |
| | 720 | 0.426 | 0.406 | 0.425 | 0.403 | 0.426 | 0.404 | 0.441 | 0.406 | 0.420 | 0.372 | 0.551 | 0.605 | 0.440 | 0.420 | 0.422 | 0.379 | 0.445 | 0.427 | 0.468 | 0.462 | 0.474 | 0.463 |
| Electricity | 96 | 0.220 | 0.130 | 0.226 | 0.134 | 0.238 | 0.145 | 0.238 | 0.139 | 0.224 | 0.131 | 0.237 | 0.140 | 0.281 | 0.201 | 0.222 | 0.129 | 0.240 | 0.148 | 0.272 | 0.168 | 0.308 | 0.193 |
| | 192 | 0.239 | 0.149 | 0.239 | 0.149 | 0.252 | 0.161 | 0.251 | 0.153 | 0.241 | 0.152 | 0.249 | 0.153 | 0.283 | 0.201 | 0.240 | 0.157 | 0.253 | 0.162 | 0.289 | 0.184 | 0.315 | 0.201 |
| | 336 | 0.254 | 0.164 | 0.257 | 0.166 | 0.267 | 0.175 | 0.266 | 0.169 | 0.248 | 0.160 | 0.267 | 0.169 | 0.298 | 0.215 | 0.259 | 0.163 | 0.269 | 0.178 | 0.300 | 0.198 | 0.329 | 0.214 |
| | 720 | 0.278 | 0.193 | 0.288 | 0.204 | 0.303 | 0.222 | 0.297 | 0.206 | 0.298 | 0.192 | 0.301 | 0.203 | 0.331 | 0.257 | 0.290 | 0.197 | 0.317 | 0.225 | 0.320 | 0.220 | 0.355 | 0.246 |
| Traffic | 96 | 0.234 | 0.357 | 0.262 | 0.394 | 0.268 | 0.407 | 0.282 | 0.388 | 0.248 | 0.362 | 0.282 | 0.410 | 0.389 | 0.649 | 0.249 | 0.360 | 0.268 | 0.395 | 0.321 | 0.593 | 0.366 | 0.587 |
| | 192 | 0.243 | 0.377 | 0.265 | 0.404 | 0.278 | 0.430 | 0.290 | 0.407 | 0.247 | 0.374 | 0.287 | 0.423 | 0.366 | 0.601 | 0.256 | 0.379 | 0.276 | 0.417 | 0.336 | 0.617 | 0.373 | 0.604 |
| | 336 | 0.250 | 0.388 | 0.270 | 0.413 | 0.281 | 0.444 | 0.294 | 0.412 | 0.271 | 0.385 | 0.296 | 0.436 | 0.369 | 0.609 | 0.264 | 0.392 | 0.283 | 0.433 | 0.336 | 0.629 | 0.383 | 0.621 |
| | 720 | 0.273 | 0.428 | 0.285 | 0.448 | 0.300 | 0.477 | 0.312 | 0.450 | 0.288 | 0.430 | 0.315 | 0.466 | 0.387 | 0.647 | 0.286 | 0.432 | 0.302 | 0.467 | 0.350 | 0.640 | 0.382 | 0.626 |
| Weather | 96 | 0.204 | 0.161 | 0.191 | 0.151 | 0.204 | 0.164 | 0.212 | 0.162 | 0.201 | 0.147 | 0.237 | 0.176 | 0.232 | 0.192 | 0.198 | 0.149 | 0.214 | 0.174 | 0.220 | 0.172 | 0.296 | 0.217 |
| | 192 | 0.246 | 0.205 | 0.232 | 0.193 | 0.250 | 0.214 | 0.248 | 0.204 | 0.234 | 0.189 | 0.282 | 0.220 | 0.271 | 0.240 | 0.241 | 0.194 | 0.254 | 0.221 | 0.261 | 0.219 | 0.336 | 0.276 |
| | 336 | 0.283 | 0.253 | 0.273 | 0.244 | 0.291 | 0.269 | 0.286 | 0.254 | 0.279 | 0.262 | 0.319 | 0.265 | 0.307 | 0.292 | 0.282 | 0.245 | 0.296 | 0.278 | 0.306 | 0.280 | 0.380 | 0.339 |
| | 720 | 0.332 | 0.322 | 0.323 | 0.311 | 0.352 | 0.355 | 0.337 | 0.326 | 0.316 | 0.304 | 0.362 | 0.323 | 0.353 | 0.364 | 0.334 | 0.314 | 0.347 | 0.358 | 0.359 | 0.365 | 0.428 | 0.403 |
| $1^{st}$ **Wins** | | **11** | | **18** | | **0** | | **1** | | **25** | | **0** | | **0** | | **4** | | **0** | | **0** | | **0** | |

Table 19: Comparison between simple ablations, LLM-based methods, and non-LLM methods. LLM-based methods slightly outperform non-LLM methods. The non-LLM results are from the LLM-Time and iTransformer papers, which are computed on the same splits. As expected, the main findings in our paper are unchanged: The LLM-based methods are slightly better, but our ablations indicate this isn't due to the LLM. Red: Best performance. Blue: Second Best.

| Methods Metric | Pre+FT (GPT2) | | woPre+FT | | Pre+woFT | | woPre+woFT | |
|---|---|---|---|---|---|---|---|---|
| | MAE | MSE | MAE | MSE | MAE | MSE | MAE | MSE |
| **ETTh1** 96 | 0.3927 | 0.3695 | 0.3925 | 0.3747 | 0.3920 | 0.3735 | 0.3969 | 0.3798 |
| 192 | 0.4258 | 0.4290 | 0.4185 | 0.4268 | 0.4178 | 0.4243 | 0.4276 | 0.4345 |
| 336 | 0.4404 | 0.4510 | 0.4397 | 0.4627 | 0.4316 | 0.4526 | 0.4413 | 0.4656 |
| 720 | 0.4661 | 0.4757 | 0.4629 | 0.4807 | 0.4654 | 0.4863 | 0.4804 | 0.5099 |
| **ETTh2** 96 | 0.3359 | 0.2841 | 0.3291 | 0.2741 | 0.3340 | 0.2829 | 0.3336 | 0.2831 |
| 192 | 0.3782 | 0.3532 | 0.3780 | 0.3526 | 0.3758 | 0.3510 | 0.3778 | 0.3547 |
| 336 | 0.3937 | 0.3611 | 0.3992 | 0.3697 | 0.3946 | 0.3649 | 0.4006 | 0.3667 |
| 720 | 0.4275 | 0.4057 | 0.4295 | 0.4067 | 0.4277 | 0.4067 | 0.4366 | 0.4172 |
| **ETTm1** 96 | 0.3497 | 0.3228 | 0.3497 | 0.3240 | 0.3483 | 0.3231 | 0.3516 | 0.3243 |
| 192 | 0.3756 | 0.3751 | 0.3783 | 0.3789 | 0.3751 | 0.3750 | 0.3775 | 0.3801 |
| 336 | 0.4009 | 0.4108 | 0.3998 | 0.4092 | 0.4014 | 0.4105 | 0.4050 | 0.4152 |
| 720 | 0.4378 | 0.4765 | 0.4454 | 0.4931 | 0.4343 | 0.4730 | 0.4456 | 0.4916 |
| **ETTm2** 96 | 0.2553 | 0.1771 | 0.2529 | 0.1752 | 0.2544 | 0.1771 | 0.2546 | 0.1768 |
| 192 | 0.3002 | 0.2446 | 0.2986 | 0.2466 | 0.2991 | 0.2451 | 0.2985 | 0.2438 |
| 336 | 0.3413 | 0.3086 | 0.3375 | 0.3064 | 0.3401 | 0.3079 | 0.3397 | 0.3070 |
| 720 | 0.3953 | 0.4023 | 0.3995 | 0.4124 | 0.3947 | 0.4005 | 0.3969 | 0.4038 |
| **Illness** 24 | 0.7876 | 1.4596 | 0.7862 | 1.4470 | 0.8327 | 1.5333 | 0.8346 | 1.5892 |
| 36 | 0.8373 | 1.5726 | 0.8317 | 1.5531 | 0.8308 | 1.5083 | 0.8438 | 1.5347 |
| 48 | 0.8895 | 1.7839 | 0.9300 | 1.8532 | 0.8931 | 1.7503 | 0.9033 | 1.7689 |
| 60 | 0.9619 | 1.9824 | 0.8612 | 1.6051 | 0.9400 | 1.8644 | 0.8837 | 1.6594 |
| **Weather** 96 | 0.2065 | 0.1675 | 0.2092 | 0.1678 | 0.2076 | 0.1677 | 0.2112 | 0.1744 |
| 192 | 0.2506 | 0.2159 | 0.2526 | 0.2172 | 0.2555 | 0.2165 | 0.2567 | 0.2247 |
| 336 | 0.2923 | 0.2709 | 0.2967 | 0.2734 | 0.2939 | 0.2714 | 0.2949 | 0.2777 |
| 720 | 0.3454 | 0.3495 | 0.3458 | 0.3494 | 0.3514 | 0.3582 | 0.3476 | 0.3559 |
| **Traffic** 96 | 0.2737 | 0.4159 | 0.2622 | 0.4103 | 0.2708 | 0.4158 | 0.2751 | 0.4201 |
| 192 | 0.2764 | 0.4302 | 0.2694 | 0.4287 | 0.2742 | 0.4324 | 0.2776 | 0.4356 |
| 336 | 0.2863 | 0.4507 | 0.2781 | 0.4456 | 0.2817 | 0.4485 | 0.2868 | 0.4530 |
| 720 | 0.3010 | 0.4783 | 0.2986 | 0.4792 | 0.3013 | 0.4817 | 0.3056 | 0.4844 |
| **Electricity** 96 | 0.2397 | 0.1469 | 0.2308 | 0.1377 | 0.2382 | 0.1442 | 0.2430 | 0.1530 |
| 192 | 0.2538 | 0.1629 | 0.2506 | 0.1581 | 0.2522 | 0.1605 | 0.2541 | 0.1652 |
| 336 | 0.2701 | 0.1785 | 0.2676 | 0.1733 | 0.2673 | 0.1768 | 0.2699 | 0.1797 |
| 720 | 0.3004 | 0.2148 | 0.2897 | 0.1987 | 0.2964 | 0.2104 | 0.2980 | 0.2156 |
| **#Wins** | 17 | | 28 | | 17 | | 2 | |

Table 20: Pretraining on language datasets is not necessary for time series forecasting tasks. The table shows the performance of using pretraining models versus not using pretraining, as well as the combination of fine-tuning and not fine-tuning LLMs in time series forecasting.

| Dataset | | | ETTh1 | | | | Illness | | | |
|---|---|---|---|---|---|---|---|---|---|---|
| | Predict Lengths | | Sf-all. | Sf-half. | Ex-half | Masking | Sf-all. | Sf-half. | Ex-half | Masking |
| | 96(24) | Time-LLM | 51.8% | 5.6% | 79.6% | 32.5% | 99.0% | 33.6% | 34.9% | 64.6% |
| | | **w/o LLM** | 56.0% | 4.5% | 89.7% | 39.5% | 76.5% | 20.9% | 18.4% | 53.0% |
| | | **LLM2Attn** | 53.8% | 3.3% | 92.2% | 33.8% | 72.7% | 20.4% | 13.1% | 44.6% |
| | | **LLM2Trsf** | 50.3% | 3.4% | 89.2% | 34.8% | 74.5% | 23.0% | 14.3% | 49.3% |
| | 192(36) | Time-LLM | 43.9% | 5.9% | 72.1% | 32.5% | 83.9% | 43.4% | 30.7% | 69.9% |
| | | **w/o LLM** | 46.8% | 4.3% | 77.3% | 32.9% | 77.0% | 24.5% | 15.3% | 56.6% |
| | | **LLM2Attn** | 45.0% | 4.4% | 78.0% | 30.0% | 73.8% | 25.7% | 11.2% | 53.3% |
| Time-LLM [15] | | **LLM2Trsf** | 44.0% | 5.5% | 77.1% | 29.9% | 72.1% | 27.0% | 6.0% | 51.3% |
| | 336(48) | Time-LLM | 39.1% | 5.9% | 67.0% | 24.7% | 51.6% | 26.0% | 11.2% | 48.5% |
| | | **w/o LLM** | 41.7% | 3.9% | 71.6% | 29.9% | 73.6% | 27.6% | 13.6% | 52.9% |
| | | **LLM2Attn** | 34.1% | 1.2% | 69.5% | 21.1% | 85.1% | 38.8% | 12.8% | 57.5% |
| | | **LLM2Trsf** | 54.9% | 3.7% | 75.0% | 26.3% | 85.2% | 42.7% | 18.6% | 63.6% |
| | 720(60) | Time-LLM | 72.6% | 31.8% | 73.3% | 32.1% | 73.0% | 42.4% | 27.5% | 76.3% |
| | | **w/o LLM** | 39.7% | 5.2% | 64.3% | 29.3% | 74.1% | 27.7% | 14.2% | 56.6% |
| | | **LLM2Attn** | 45.5% | 6.0% | 66.0% | 26.0% | 66.2% | 28.5% | 11.9% | 51.7% |
| | | **LLM2Trsf** | 53.3% | 10.2% | 63.7% | 27.3% | 67.7% | 30.7% | 7.8% | 49.2% |

| Dataset | | | ETTh1 | | | | Illness | | | |
|---|---|---|---|---|---|---|---|---|---|---|
| | Predict Lengths | | Sf-all. | Sf-half. | Ex-half | Masking | Sf-all. | Sf-half. | Ex-half | Masking |
| | 96(24) | CALF | 50.5% | 9.6% | 5.6% | 8.5% | 113.0% | 47.4% | 24.4% | 22.9% |
| | | **w/o LLM** | 56.2% | 12.1% | 6.1% | 10.4% | 118.0% | 50.4% | 45.8% | 28.9% |
| | | **LLM2Attn** | 51.9% | 10.8% | 5.8% | 7.3% | 87.3% | 42.4% | 35.1% | 25.8% |
| | | **LLM2Trsf** | 50.3% | 8.5% | 5.5% | 7.0% | 102.6% | 56.2% | 32.6% | 26.0% |
| | 192(36) | CALF | 41.7% | 7.8% | 3.3% | 3.6% | 100.9% | 45.7% | 12.9% | 17.1% |
| | | **w/o LLM** | 50.9% | 13.5% | 4.5% | 6.0% | 115.6% | 57.0% | 28.3% | 21.4% |
| | | **LLM2Attn** | 45.8% | 9.7% | 3.6% | 5.0% | 78.8% | 41.8% | 10.1% | 19.9% |
| CALF [22] | | **LLM2Trsf** | 42.4% | 8.3% | 3.4% | 4.3% | 73.6% | 42.5% | 7.0% | 17.5% |
| | 336(48) | CALF | 38.1% | 9.0% | 1.7% | 5.0% | 68.9% | 43.0% | 15.1% | 14.9% |
| | | **w/o LLM** | 47.2% | 14.5% | 2.3% | 8.7% | 76.3% | 49.0% | 18.5% | 18.1% |
| | | **LLM2Attn** | 39.3% | 10.0% | 1.7% | 5.5% | 78.9% | 54.2% | 22.2% | 15.3% |
| | | **LLM2Trsf** | 40.3% | 10.0% | 1.9% | 4.9% | 78.9% | 52.0% | 23.4% | 17.7% |
| | 720(60) | CALF | 36.5% | 10.0% | 0.7% | 5.5% | 63.8% | 22.7% | 5.4% | 15.3% |
| | | **w/o LLM** | 41.0% | 10.2% | 1.2% | 6.2% | 69.7% | 30.3% | 12.1% | 19.4% |
| | | **LLM2Attn** | 36.0% | 9.9% | 0.7% | 5.0% | 71.7% | 29.2% | 12.7% | 14.2% |
| | | **LLM2Trsf** | 35.2% | 9.3% | 0.7% | 5.1% | 90.4% | 55.3% | 26.4% | 21.9% |

| Dataset | | | ETTh1 | | | | Illness | | | |
|---|---|---|---|---|---|---|---|---|---|---|
| | Predict Lengths | | Sf-all. | Sf-half. | Ex-half | Masking | Sf-all. | Sf-half. | Ex-half | Masking |
| | 96(24) | OneFitsAll | 62.1% | 6.1% | 16.6% | 31.3% | 86.2% | 30.9% | 36.7% | 77.5% |
| | | **w/o LLM** | 58.6% | 6.1% | 19.2% | 36.1% | 68.9% | 13.0% | 17.3% | 43.5% |
| | | **LLM2Attn** | 68.5% | 9.0% | 15.0% | 34.4% | 108.3% | 39.8% | 44.2% | 74.2% |
| | | **LLM2Trsf** | 58.0% | 7.8% | 12.6% | 30.2% | 90.8% | 27.4% | 40.3% | 60.6% |
| | 192(36) | OneFitsAll | 52.7% | 8.2% | 8.8% | 28.5% | 52.0% | 15.2% | 8.5% | 44.4% |
| | | **w/o LLM** | 47.5% | 6.1% | 10.6% | 31.8% | 76.3% | 24.0% | 18.0% | 47.5% |
| | | **LLM2Attn** | 80.8% | 13.4% | 6.7% | 25.6% | 97.8% | 42.0% | 36.3% | 65.0% |
| OneFitsAll [50] | | **LLM2Trsf** | 54.7% | 12.5% | 6.4% | 24.1% | 82.0% | 29.6% | 24.4% | 63.8% |
| | 336(48) | OneFitsAll | 62.1% | 6.1% | 16.6% | 31.3% | 79.5% | 34.8% | 25.1% | 74.5% |
| | | **w/o LLM** | 58.6% | 6.1% | 19.2% | 36.1% | 78.1% | 22.1% | 15.7% | 49.4% |
| | | **LLM2Attn** | 68.5% | 9.0% | 15.0% | 34.4% | 86.4% | 45.8% | 21.7% | 69.8% |
| | | **LLM2Trsf** | 58.0% | 7.8% | 12.6% | 30.2% | 89.2% | 35.8% | 21.6% | 66.9% |
| | 720(60) | OneFitsAll | 40.1% | 8.0% | 3.5% | 25.4% | 53.6% | 21.8% | 9.9% | 44.9% |
| | | **w/o LLM** | 39.4% | 7.1% | 4.1% | 28.2% | 74.5% | 20.9% | 17.1% | 45.5% |
| | | **LLM2Attn** | 93.8% | 21.9% | 1.5% | 25.3% | 86.0% | 41.4% | 27.0% | 72.8% |
| | | **LLM2Trsf** | 87.2% | 20.1% | 1.6% | 27.1% | 92.3% | 40.7% | 29.3% | 76.3% |

Table 21: Results for input shuffling/masking for Time-LLM, CALF, and OneFitsAll methods on ETTh1 (predict length are "96, 192, 336 and 720") and Illness (predict length are "24, 36, 48 and 60"), the impact of shuffling the input on the degradation of time series forecasting performance does not change significantly before and after model modifications.

| Dataset | ETTh2 | | | | Electricity | | | |
| --- | --- | --- | --- | --- | --- | --- | --- | --- |
| Input Ablation | Sf-all. | Sf-half. | Ex-half | Masking | Sf-all. | Sf-half. | Ex-half | Masking |
| Time-LLM | 27.1% | 4.5% | 44.6% | 99.5% | 212.1% | 52.2% | 323.4% | 103.3% |
| **w/o LLM** | 31.2% | 2.9% | 52.4% | 137.9% | 220.9% | 47.3% | 332.4% | 105.5% |
| **LLM2Attn** | 30.7% | 2.9% | 46.7% | 134.8% | 234.7% | 58.2% | 350.4% | 115.3% |
| **LLM2Trsf** | 24.8% | 2.3% | 43.4% | 115.0% | 240.8% | 50.8% | 362.0% | 118.5% |
| OneFitsAll | 33.0% | 1.1% | 39.7% | 112.1% | 237.8% | 25.2% | 382.0% | 88.9% |
| **w/o LLM** | 32.7% | 2.8% | 38.7% | 115.8% | 249.7% | 43.4% | 384.7% | 109.2% |
| **LLM2Attn** | 27.5% | 1.0% | 36.5% | 137.4% | 255.2% | 49.5% | 393.9% | 119.2% |
| **LLM2Trsf** | 29.6% | 2.2% | 37.2% | 130.8% | 263.8% | 55.1% | 406.4% | 125.4% |
| CALF | 11.3% | 2.0% | 9.1% | 44.7% | 235.1% | 108.2% | 29.3% | 37.6% |
| **w/o LLM** | 23.1% | 4.1% | 17.2% | 46.9% | 239.5% | 109.2% | 28.6% | 39.4% |
| **LLM2Attn** | 16.7% | 3.4% | 14.8% | 31.7% | 237.9% | 116.2% | 28.9% | 42.2% |
| **LLM2Trsf** | 18.9% | 3.0% | 15.3% | 43.2% | 245.6% | 107.0% | 29.2% | 42.3% |

Table 22: For the input shuffling/masking experiments on ETTh2 and Electricity, the impact of shuffling the input on the degradation of time series forecasting performance does not change significantly before and after model modifications.

| Dataset | ETTm1 | | | | ETTm2 | | | |
| --- | --- | --- | --- | --- | --- | --- | --- | --- |
| Input Ablation | Sf-all. | Sf-half. | Ex-half | Masking | Sf-all. | Sf-half. | Ex-half | Masking |
| Time-LLM | 66.6% | 10.1% | 107.7% | 43.5% | 47.0% | 5.3% | 77.7% | 202.9% |
| **w/o LLM** | 68.7% | 12.4% | 112.0% | 52.6% | 49.8% | 4.7% | 78.7% | 199.6% |
| **LLM2Attn** | 73.5% | 13.5% | 119.3% | 50.7% | 49.6% | 4.7% | 75.9% | 196.3% |
| **LLM2Trsf** | 72.0% | 27.6% | 117.0% | 54.2% | 46.4% | 3.7% | 76.5% | 191.9% |
| OneFitsAll | 74.4% | 8.0% | 123.4% | 41.8% | 48.0% | 3.7% | 82.9% | 172.9% |
| **w/o LLM** | 66.7% | 10.3% | 115.1% | 45.0% | 48.1% | 4.4% | 81.9% | 190.7% |
| **LLM2Attn** | 73.9% | 11.1% | 124.8% | 51.6% | 45.5% | 3.3% | 77.8% | 183.6% |
| **LLM2Trsf** | 70.1% | 8.3% | 126.2% | 50.5% | 48.2% | 4.3% | 82.0% | 182.9% |
| CALF | 64.3% | 24.8% | 122.3% | 13.7% | 23.5% | 9.1% | 47.1% | 61.7% |
| **w/o LLM** | 66.7% | 26.0% | 123.6% | 15.9% | 27.1% | 11.2% | 51.9% | 58.1% |
| **LLM2Attn** | 62.5% | 23.4% | 122.3% | 15.6% | 25.2% | 8.8% | 51.5% | 57.3% |
| **LLM2Trsf** | 61.8% | 25.6% | 122.6% | 13.6% | 23.7% | 8.2% | 51.0% | 59.3% |

Table 23: For the input shuffling/masking experiments on ETTm1 and ETTm2, the impact of shuffling the input on the degradation of time series forecasting performance does not change significantly before and after model modifications.

| Dataset | Weather | | | | Traffic | | | |
| --- | --- | --- | --- | --- | --- | --- | --- | --- |
| Input Ablation | Sf-all. | Sf-half. | Ex-half | Masking | Sf-all. | Sf-half. | Ex-half | Masking |
| Time-LLM | 65.8% | 5.5% | 85.9% | 85.9% | 198.1% | 56.6% | 309.8% | 101.3% |
| **w/o LLM** | 59.2% | 4.7% | 78.7% | 74.6% | 196.8% | 70.2% | 282.1% | 78.3% |
| **LLM2Attn** | 71.8% | 8.5% | 91.6% | 95.3% | 212.3% | 82.4% | 312.7% | 92.2% |
| **LLM2Trsf** | 71.9% | 7.1% | 94.9% | 103.2% | 197.5% | 64.6% | 307.2% | 101.1% |
| OneFitsAll | 71.0% | 9.8% | 102.1% | 97.7% | 206.4% | 11.7% | 369.1% | 93.5% |
| **w/o LLM** | 56.4% | 2.3% | 81.2% | 73.5% | 238.4% | 67.0% | 338.6% | 85.9% |
| **LLM2Attn** | 64.9% | 5.4% | 90.2% | 94.8% | 222.6% | 57.5% | 350.5% | 95.1% |
| **LLM2Trsf** | 77.8% | 8.3% | 101.9% | 111.0% | 217.8% | 35.9% | 361.3% | 113.3% |
| CALF | 32.6% | 4.2% | 54.5% | 27.2% | 201.7% | 105.0% | 63.1% | 27.2% |
| **w/o LLM** | 28.1% | 5.9% | 49.4% | 13.6% | 217.3% | 115.4% | 67.2% | 29.7% |
| **LLM2Attn** | 26.0% | 6.2% | 49.2% | 17.1% | 226.7% | 122.7% | 69.4% | 30.7% |
| **LLM2Trsf** | 28.0% | 6.4% | 54.1% | 21.1% | 232.8% | 120.5% | 72.0% | 29.6% |

Table 24: For the input shuffling/masking experiments on Weather and Traffic, the impact of shuffling the input on the degradation of time series forecasting performance does not change significantly before and after model modifications.

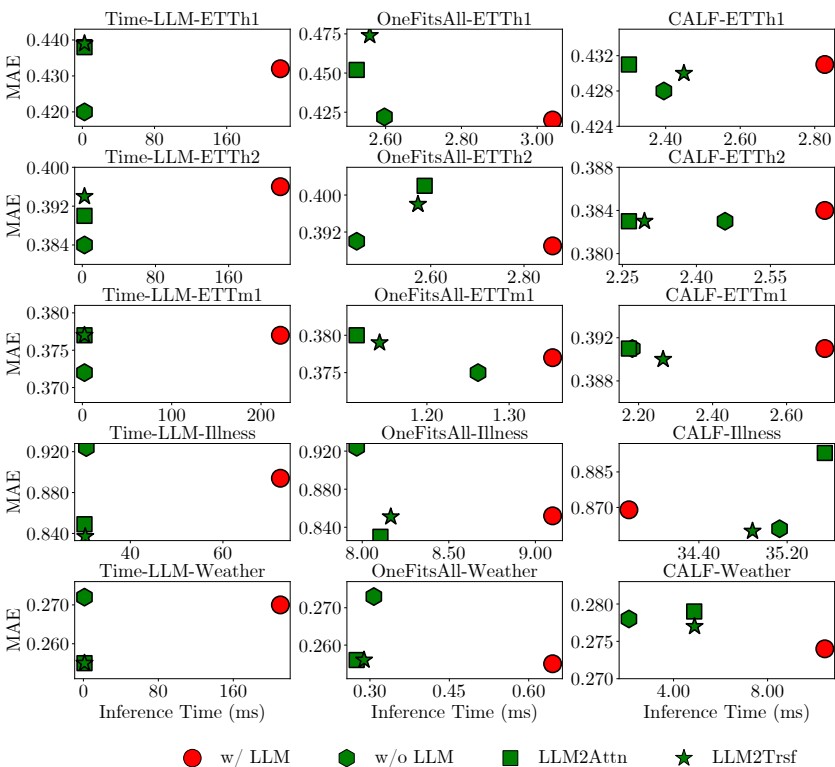

Figure 7: Ablation methods consume less time for inference while providing better forecasting performance in most cases. The figure above shows the inference time and prediction accuracy of Time-LLM, OneFitsAll, and CALF on ETTh1, ETTh2, ETTm1, Illness, and Weather, Traffic datasets, averaged across prediction lengths. Results of other datasets refer to Figure 3.

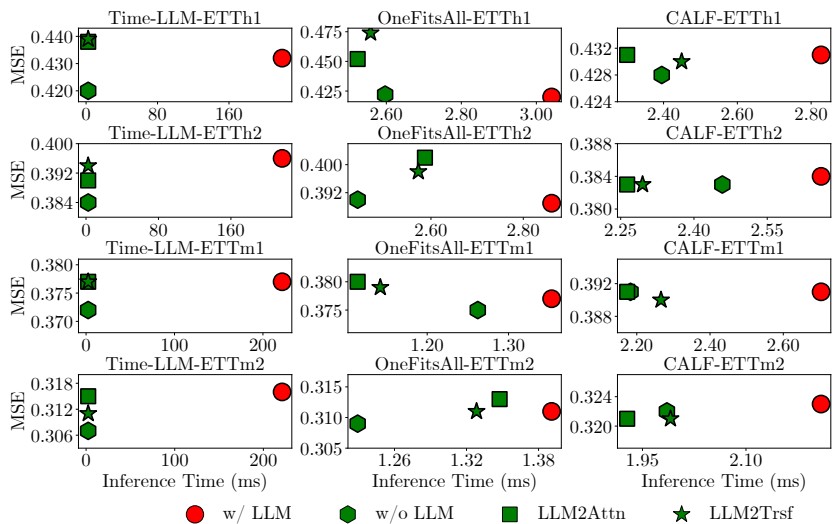

(a) Inference Time and Performance for ETTh1, ETTh2, ETTm1, and ETTm2.

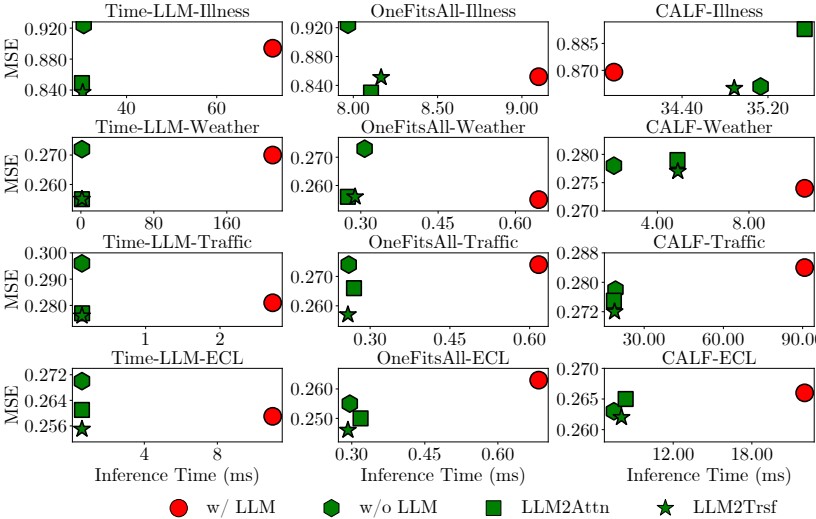

(b) Inference Time and Performance for Illness, Weather, Traffic, and Electricity.

Figure 8: Ablation methods consume less time for inference while providing better forecasting performance in most cases. The figure above shows the inference time and prediction accuracy of Time-LLM, OneFitsAll, and CALF on all the datasets, averaged across prediction lengths in MSE metric.

