# OpenReview forum: "Are Language Models Actually Useful for Time Series Forecasting?"
_NeurIPS.cc/2024/Conference — NeurIPS 2024 spotlight_

### Official Review · Reviewer_CG3z · 2024-06-29

**Soundness:** 3
**Presentation:** 3
**Contribution:** 3
**Rating:** 7
**Confidence:** 4

**Summary:**

This paper questions the effectiveness of LLMs in time series forecasting. Through a series of ablation studies on 3 LLM-based forecasting methods, the authors find that removing or replacing the LLM component often improves forecasting results. They conclude that LLMs do not provide significant benefits for time series forecasting tasks, despite their significant computational cost. The paper suggests that simpler models or encoders with basic attention mechanisms perform equally well, challenging the current trend of employing LLMs for time series forecasting.

**Strengths:**

- The paper offers an interesting perspective on the use of LLMs in time series forecasting, challenging prevalent assumptions and trends in the field.
- The empirical analysis seems to be valid, with well-executed ablation studies and comparisons across multiple methods and datasets.
- The paper is easy to read, with well-organized sections and effective use of figures and tables to present relevant results.
- The findings may have implications for the research community, potentially redirecting efforts towards more efficient and effective methods for time series forecasting.

**Weaknesses:**

- The scope is limited to time series forecasting and does not explore other potential applications of LLMs in time series analysis, such as imputation, classification or anomaly detection.
- The datasets used are all evenly spaced time series, which may not fully represent the variety of real-world time series data, such as those with irregular intervals.
- The paper could benefit from a more detailed discussion on the potential reasons why LLMs fail to outperform simpler models in this context, providing deeper theoretical insights.

**Questions:**

Related to my last point in "weakness", can the authors provide more theoretical explanations or hypotheses on why LLMs do not perform well in time series forecasting compared to simpler models?

**Limitations:**

The authors acknowledge that the study focuses solely on time series forecasting and uses only evenly spaced datasets. Future research should explore LLMs’ effectiveness in other time series tasks and with non-uniform datasets. Additionally, potential negative societal impacts, such as the environmental cost of training large models, should be considered.

---

> ### Author Rebuttal · Authors · 2024-08-06
>
> Thanks for your thoughtful and positive review! We've responded to each of your points below.
>
> **W1: Can our work extend to other uses of LLMs for time series?**
> We agree this is an exciting, natural direction for our work. While it’s beyond the scope of one paper, we hope our work inspires the community to do exactly this, ultimately taking a closer look at such LLM-based methods broadly.
>
> **W2: Can our work extend to irregularly-sampled time series?**
> Thanks for noting our focus on evenly-spaced time series. We completely agree that irregularly-sampled time series are of critical interest and we hope to consider them in future works. For this current work, we believe our hypotheses are best tested by following the prior works’ experimental setups as closely as possible and as LLMs are used for more data types, we hope they are subjected to careful examination!
>
> **W3 and Question: Could we provide deeper discussions of LLMs’ weaknesses and give theoretical insights?**
> This is a great idea and diving deeper into *why* the LLMs underperform is extremely interesting. But there are many open questions about this type of interpretability and many challenges of proving negative results. While we probe at this “why” question in Sections 4.3-4.6, we hope our work inspires others to ask similar questions and join in developing the theory behind when/why/if LLMs can benefit time series forecasting (and other tasks). As it stands, we believe our work successfully presents tension between the promise of big, multi-modal models and a need to deeply understand the sources of practical performance.
>
> Per your direct question, developing theoretical explanations should be done with care over longer periods of time than a rebuttal. But in the spirit of brainstorming, we provide two speculative lines of attack for future works:
> 1. There may be a lack of key, transferrable properties of time series in LLM pre-training data. For example, forecasting often depends on learning periodic trends in time series, which are not a key signal present in natural language.
> 2. Long, forecastable sequences of numbers may not be in the pretraining data, driving a need to update an LLM’s weights beyond the point of leveraging its language abilities.

---

### Official Review · Reviewer_wccd · 2024-07-05

**Soundness:** 4
**Presentation:** 4
**Contribution:** 4
**Rating:** 8
**Confidence:** 4

**Summary:**

A recent surge of papers have popularised the usage of pre-trained LLMs for time series forecasting. The paper analyses the claim that LLMs are useful for time series forecasting, by performing a series of ablation studies. Their conclusion is that LLMs bring little to no benefit for the task, and are significantly more costly.

**Strengths:**

1. The paper is very well written, the hypothesis is stated clearly and the experiments and results are also clearly stated.
2. Experimental approach is very sound, experiments are well thought out and very comprehensive.
3. The paper reveals very significant findings regarding a recent trend in time series forecasting, in which many papers at top conferences have been publishing about. It turns out that the performance of many of these methods is not due to a pre-trained LLM as touted by these papers, but due to other factors such as patching and channel independence is the most important.

**Weaknesses:**

1. Experiments in section 4.3 are only presented for LLaTa, it should be presented for all the models. Importantly, the OneFitsAll paper showed that pre-trained weights performed much better than random initialized weights. Can the authors comment on this?

2. The experiments in 4.6 lead to a model that is very similar to PatchTST and some other similar patch-based Transformers, which the LLM-based models claim to be better than. Can the authors comment on this?

**Questions:**

See weaknesses

**Limitations:**

Authors have adequately addressed limitations in appendix.

---

> ### Author Rebuttal · Authors · 2024-08-06
>
> Thank you for the positive and encouraging feedback and for your endorsement of our work! We've addressed your questions below.
>
> **W1: Why does RQ3 only consider LLaTa?**
>
> Thanks for suggesting we extend RQ3 to include all methods. We completely agree this would strengthen this RQ and our choice was due to computational constraints—**The “woPre+FT” ablation requires training the LLM from scratch**. Training Time-LLM’s 7B base LLM so many times is beyond our immediate resources. But we will take your suggestion and see if we can find a way to run this for the final version. We also omitted OneFitsAll from this experiment because OneFitsAll and LLaTA have similar performance, use the same base LLM, and our aim in this RQ is just on the impact of the LLM’s pretraining. But we will take your suggestion and run this experiment. We strongly believe this finding doesn't hinge on this experiment, but we will include at least one more method (hopefully two).
>
> For OneFitsAll’s random initialization experiment, we’d like to note that their results were reported in a few-shot setting, unlike ours. Running our experiment again in a few-shot setting is an interesting future experiment, though.
>
> **W2: How does PAttn compare to PatchTST?**
>
> This is a great question! We agree that they are similar, and we’d like to note that PAttn is designed only to explore the role encoders play in LLM-based forecasters’ performance. As shown in our extended comparisons (see our uploaded pdf), PAttn and PatchTST actually do perform similarly, likely due to their architectural similarity. The key difference is PAttn lacks a position embedding and feedforward layer in the transformer.

---

### Official Review · Reviewer_UHrk · 2024-07-06

**Soundness:** 2
**Presentation:** 3
**Contribution:** 2
**Rating:** 6
**Confidence:** 4

**Summary:**

This paper explores the effectiveness of language models in time series forecasting. The authors substantiate their claim by performing three straightforward ablations of three popular and recent LLM-based forecasting methods. After extensive experiments, the authors find that patching and attention structures perform similarly to state-of-the-art LLM-based forecasters.

**Strengths:**

1. The paper is well-structured and easy to follow.
2. The author question existing LLM-based forecasting models by conducting comprehensive ablation studies and provide very interesting and insightful observations.

**Weaknesses:**

1. Although the paper focuses on the effectiveness of LLM in TSF, existing state-of-the-art forecasting models have been omitted.
2. The authors conduct ablation studies on the patching and decomposition of LLM-based models to find out where the performance comes from. However, there is a lack of discussion on LLaTA, which embeds the input time series by treating each channel as a token instead of using patching.

**Questions:**

1. Why the author choose ETTh1 and ILI datasets in RQ4? Results on other benchmark datasets are expected.
2. How does the author implement the "LTrsf" model, and what is the difference between "LTrsf" and the existing iTransformer model?
3. Can you compare the results between "PAttn" and PatchTST?

**Limitations:**

yes

---

> ### Author Rebuttal · Authors · 2024-08-06
>
> Thank you for your focused and actionable review. We are glad you agree our work provides interesting and insightful observations! As detailed point-by-point below, **we’ve addressed your remaining concerns by running your suggested experiments**.
>
> **W1: Including state-of-the-art forecasting models**
>
> Thank you for this suggestion! We’d like to clarify that **each LLM-based method claims to be state-of-the-art,** reporting they outperform non-LLM methods like PatchTST and iTransformer:
> * *“TIME-LLM is a powerful time series learner that outperforms state-of-the-art, specialized forecasting models”* (taken from Time-LLM’s abstract).
> * *“[LLaTA] establishes state-of-the-art performance for both long-term and short-term forecasting tasks”* (taken from LLaTA’s abstract; note that “LLaTA” was renamed to “CALF, which we will update in our final version).
> * *“pre-trained models on natural language or images can lead to a comparable or state-of-the-art performance in all main time series analysis tasks”* (taken from OneFitsAll’s abstract).
>
> So the merit of our work doesn’t depend on this comparison. Our reproduced MSE and MAE values are also nearly-identical to those of each method’s original experimental results, so we have no reason to believe this comparison won’t also hold. We also agree with you that comparing to non-LLM methods is largely out-of-scope of our contributions.
>
> But to address your suggestion directly, and in case future readers have a similar question, we have added extended versions of all comparison tables to the Appendix (see pdf uploaded with our rebuttal). These tables include results from the methods compared in the iTransformer paper. As expected, our main findings are unchanged: The LLM-based methods are slightly better than the non-LLM methods, but our ablations indicate this isn’t due to the LLM.
>
> **W2: Including more details on LLaTA’s source of performance**
>
> Thanks for this suggestion, we believe you are describing RQ6 where we show that a simple patching method is surprisingly performant (compared to all methods and also other new and simple baselines). While RQ6 focuses on encoders and particularly on the more-popular patching and decomposition, there are still some insights about LLaTA’s encoder. In Section 4.6, we will clarify that “LTrsf” is LLaTA’s encoder without cross-modal attention.
>
> **Q1: Why does RQ4 focus on the ETTh1 and Illness datasets?**
>
> Thank you for suggesting we add more datasets to RQ4—We chose ETTh1 and Illness randomly due to compute constraints leading up to the submission.
>
> Per your suggestion, **we ran this experiment, so now RQ4 includes all 8 datasets**. We've previewed our findings below, but could only include "Sf-all" and "Ex-half" due to OpenReview's space constraints. All results will be added to the Appendix. We also note that **answering RQ4 doesn't depend on the number of datasets** because if LLMs’ strong sequence modeling really drove their forecasting performance, shuffling time series should always drop their performance more than their ablations’.
>
> |                      | **ETTh2** |         | **Electricity** |         | **ETTm1** |         | **ETTm2** |         | **Weather** |         | **Traffic** |         |
> |:--------------:|:---------:|:-------:|:---------------:|:-------:|:---------:|:-------:|:---------:|:-------:|:-----------:|:-------:|:-----------:|:-------:|
> |  |  Sf-all.  | Ex-half |     Sf-all.     | Ex-half |  Sf-all.  | Ex-half |   Sf-all. | Ex-half |   Sf-all.   | Ex-half |   Sf-all.   | Ex-half |
> | Time-LLM       |   27.1%   |  44.6% |     212.1%      | 323.4%  |   66.6%   | 107.7%  |  47.0%    |  77.7%  |    65.8%    |  85.9%  |    198.1%   |  309.8% |
> | w/o LLM        |   31.2%   |  52.4%  |     220.9%      | 332.4%  |   68.7%   | 112.0%  |  49.8%    |  78.7%  |    59.2%    |  78.7%  |    196.8%   |  282.1% |
> | LLM2Attn       |   30.7%   |  46.7%  |     234.7%      | 350.4%  |   73.5%   | 119.3%  |  49.6%    |  75.9%  |    71.8%    |  91.6%  |    212.3%   |  312.7% |
> | LLM2Trsf       |   24.8%   |  43.4%  |     240.8%      | 362.0%  |   72.0%   | 117.0%  |  46.4%    |  76.5%  |    71.9%    |  94.9%  |    197.5%   |  307.2% |
>
>
> **Table**: Subset of results from RQ4 experiments on six more datasets. Due to OpenReview's character constraints, we only show "Sf-all" and "Ex-half" results for Time-LLM—These experiments are already completed for all methods. These new results agree with those from our original paper.
>
>
> **Q2: How is LTrsf implemented and how does it connect to iTransformer?**
>
> Thank you for noting the similarity between LTrsf and iTransformer. The key difference is that "LTrsf" leaves each channel independent, ignoring multivariate relationships. LTrsf is implemented as LLaTA's encoder with cross-modal attention removed (so it's only the transformer encoder) followed by one linear layer. We will mention this in RQ6 and clarify that they outperform one another in different cases, so their strengths appear orthogonal.
>
> **Q3 : Can we compare PAttn and PatchTST?**
>
> Thanks for suggesting we add PatchTST's results to those of PAttn. We have added this comparison to the pdf uploaded with our rebuttal. We’d like to note that comparing non-LLM forecasting methods is unrelated to our work. Still, we will add this result, per your suggestion, to establish how PAttn compares to others. Please also note that we use PAttn to probe the behavior of LLM-based forecasting methods, not to outperform non-LLM methods. However, it is competitive in many cases.

---

> > ### Comment · Reviewer_UHrk · 2024-08-11
> >
> > Thanks for your response. The rebuttal addressed my concerns. I have raised my score to 6.

---

### Official Review · Reviewer_jLi6 · 2024-07-10

**Soundness:** 4
**Presentation:** 4
**Contribution:** 4
**Rating:** 8
**Confidence:** 4

**Summary:**

This paper presents an extensive empirical study on the effect of pre-trained LLMs in time series forecasting tasks. By ablating popular LLM adaptations on widely adopted time series benchmarks, experiments in the paper show LLMs do not benefit from pre-training on text data to gain improvement in forecasting performance. It suggests that the patching and first layer of attention makes most of contribution in forecasting setups.

**Strengths:**

This paper is very well written and populated with extensive empirical evidences to support its main arguments.
1. The experiment setting covers various types of LLM adaptation to forecasting, including simple fine-tuning, PEFT and modality alignment, as well as popular time series forecasting benchmarks, which makes the conclusions fairly solid.
2. The reasoning behind the 6 research questions are convincing with rigorous ablation studies, and it covers multiple aspects of applicability of LLM in forecasting, from efficacy to efficiency.
3.  Great reproducibility evidences are provided for an empirical study.

**Weaknesses:**

A questionable piece of results is the random baseline in Table 5. It is counter-intuitive that, while the `woPre+woFT` variant is completely unfitted according to line 208-210, but its metrics are quite close to other variants in all experiments.

Minor typos/errors:
1. line 76: One method ~this~ is

**Questions:**

See weakness

**Limitations:**

Limitations are pointed out in the paper by the authors.

---

> ### Author Rebuttal · Authors · 2024-08-06
>
> Thank you for the encouraging feedback and the very positive review! We've fixed the typo and responded to your query about Table 5 below.
>
> **Why does "woPre+woFT" perform well in Table 5?**
>
> This is a great observation! We were initially surprised by this, too. But we’d like to clarify that “woPre+woFT” describes only the LLM in the model, where the LLM’s parameters are randomized (woPre) and then frozen and left untrained (woFT). **The rest of the model is still finetuned.** So the fact that its metrics are similar to the other variants actually backs up our main claim: If the rest of the architecture can achieve similar performance even with a random, frozen LLM injected into the middle, the LLM is likely not driving the forecasting performance.

---

### Author Rebuttal · Authors · 2024-08-06

Thank you to all reviewers for your thoughtful feedback—we are thrilled to see such positive reception!

To summarize this feedback, the reviewers emphasize our study's importance, noting we present **"very significant findings regarding a recent trend in time series forecasting"** [wccd], **"very interesting and insightful observations"** [UHrk], and how our **"findings may have implications for the research community, potentially redirecting efforts towards more efficient and effective methods for time series forecasting"** [CG3z]. Reviewers also recognize the quality of our experiments, commending how our paper includes **"extensive empirical evidences to support its main arguments"** [jLi6], how we show **"comprehensive ablation studies"** [UHrk], how our **"experimental approach is very sound, experiments are well thought out and very comprehensive"** [wccd], and how our **"empirical analysis seems to be valid, with well-executed ablation studies and comparisons across multiple methods and datasets"** [CG3z]. All reviewers also appreciated our clarity, noting our paper is **"very well written"** [jLi6, wccd], **"well-structured and easy to follow"** [UHrk], and **"easy to read, with well-organized sections and effective use of figures and tables to present relevant results"** [CG3z].

We’ve responded to each reviewer's questions below, and we'd also like to highlight a few general improvements we’ve made to our work based on your comments:
* Based on suggestions by Reviewer UHrk, we have run our RQ4 experiments to include **six more datasets**. This rounds out RQ4, which now includes all 8 datasets studied in our original submission. We find the same trends as those on the original 2 datasets studied in RQ2.
* Based on suggestions by Reviewers UHrk and wccd, we have **included comparisons to non-LLM methods** to our main results as a new table in the Appendix (shown in the attached pdf).
* We also ran our main experiments on **five more forecasting datasets** (ExchangeRate, Covid Deaths, Taxi, NN5, and FRED-MD). We will include these new results in the final version of our paper. We find the same trends as those on the original 8 datasets.

Results from each new experiment strengthen our findings even further. So thank you again for these suggestions, they have concretely improved this work!

---

### Decision · Program_Chairs · 2024-09-25

**Decision:**

Accept (spotlight)

**Comment:**

The paper presents an interesting empirical study by questioning the effectiveness of LLMs in time series forecasting.  The study considers ablations of LLM adaptation to forecasting and popular time series forecasting benchmarks. The finding that pretrained LLMs are not better than models trained from scratch is worth publishing.